# Hyperspectral anomaly detection leveraging spatial attention and right-shifted spectral energy

**Ruhan A**[1], **Quanxue Gao**[2]*, **Xiaoni Zhang**[3], **Wenwen Feng**[4], with and **Siti Khadijah Ali**[4]

**1** Xianyang Normal University, Xianyang, Shaanxi, China, **2** Xidian University, Xi'an, Shaanxi, China, **3** Xi'an Peihua University, Xi'an, Shaanxi, China, **4** Universiti Putra Malaysia, Serdang, Malaysia

* qxgao@xidian.edu.cn

**Data availability statement:** All data may be found in the following public repository: https://earthexplorer.usgs.gov/.

## Abstract

In this research, we have proposed a novel anomaly detection algorithm for processing hyperspectral images (HSIs), called the Graph Attention Network–Beta Wavelet Graph Neural Network-based Hyperspectral Anomaly Detection (GAN–BWGNN HAD). This algorithm treats each pixel as a node in a graph, where edges represent pixel correlations and node attributes correspond to spectral features. The algorithm integrates spatial and spectral information, utilizing graph neural networks to identify nonlinear relationships within the image, thereby enhancing anomaly detection precision. The K-nearest neighbor (KNN) algorithm facilitates the creation of edges between pixels, enabling the incorporation of distant pixels and improving resilience to noise and local irregularities. The GAN component incorporates an adaptive attention mechanism to dynamically prioritize relevant spatial features. The BWGNN component employs beta wavelets as a localized bandpass filter, effectively identifying spectral anomalies by addressing the right-shifted spectral energy phenomenon. Furthermore, the utilization of beta wavelets obviates the necessity for computationally intensive Laplacian matrix decompositions, thereby enhancing processing efficiency. This approach effectively integrates spatial and spectral information, providing a more accurate and efficient solution for hyperspectral anomaly detection. Experiments on six real-world hyperspectral datasets and one simulated dataset demonstrate the superior performance of our proposed method. It consistently achieved high Area Under the Curve (AUC) values (e.g., 0.9986 on AVIRIS-II, 0.9961 on abu-beach-2, 0.9982 on abu-urban-3, 0.9999 on Salinas-simulate, 0.9872 on Cri), significantly outperforming state-of-the-art methods. The proposed method also exhibited sub-second detection times (0.20–0.28 s) on most datasets, significantly faster than traditional methods (achieving a speedup of 100 to 500 times) and deep learning models (achieving a speedup of 6 to 8 times).

**Funding:** This research was funded by the Xi'an Peihua University Research Institutions and Innovation Team Special Project under Grant PHJT2406. The funders had no role in study design, data collection and analysis, decision to publish, or preparation of the manuscript.

**Competing interests:** The authors declare that they have no known competing financial interests or personal relationships that could have appeared to influence the work reported in this paper.

## Introduction

Hyperspectral remote sensing captures detailed spectral information across hundreds of contiguous narrow bands, typically spanning the visible to infrared spectrum (400–2500 nm). This generates a three-dimensional (3D) hyperspectral image (HSI) data cube, where each pixel contains a high-resolution spectral signature, enabling fine-grained material identification beyond the capabilities of traditional RGB or multispectral imaging. These capabilities make HSIs invaluable for diverse applications including mineral exploration, precision agriculture, environmental monitoring, and military surveillance, where detecting rare or unexpected targets—known as hyperspectral anomaly detection (HAD)—is frequently a critical task.

Hyperspectral anomaly detection (HAD) techniques can be broadly categorized into four main types: statistical models, collaborative representations, low-rank and sparse decomposition, and deep learning [1–12]. Statistical models, such as the RX algorithm [13], detect anomalies by computing the Mahalanobis distance between a test pixel and the background mean. Variants like global RX (GRX) and local RX (LRX) improve detection accuracy by estimating background statistics using the entire image or local regions. However, these methods are computationally intensive, sensitive to data distribution assumptions, and struggle with non-Gaussian distributed data [14]. They also rely heavily on surrounding pixel data, making them susceptible to noise and anomalies. Recent advances like Information Entropy Estimation based on Point-Set Topology (IEEPST) [15] and the Chessboard Topology-based Anomaly Detection (CTAD) algorithm [16] address some of these limitations. IEEPST maps hyperspectral image data into ordered topological spaces, revealing the data's mathematical-statistical properties and addressing data-model discrepancies. CTAD decomposes hyperspectral images using a chessboard-shaped framework to extract deep-level features, quantify differences between anomalies and background, and highlight spectral trends. These methods, free from specific model assumptions, adaptively learn data features, achieving robust and generalizable anomaly detection results. However, they still face challenges in efficiently integrating spatial and spectral information, handling non-Gaussian data, and being resilient to noise and anomalies.

The collaborative representation detection (CRD) algorithm is a classic anomaly detection algorithm proposed by Li and Du. This algorithm relies on the linear representation of background pixels using surrounding data [17]. Establishing dual windows and calculating background information for each pixel is computationally expensive for large-scale hyper spectral data. Moreover, CRD utilizing linear representation methods may struggle to adequately manage and characterize nonlinear relationships.

Low-rank and sparse decomposition techniques [18,19], such as the Hyperspectral Anomaly Detection via Generalized Shrinkage Mappings (HADGSM) [20], have been proposed to capture spectral correlations, spatial smoothness, and local geometrical structures in hyperspectral images. These methods leverage nonconvex penalties for group sparsity, $l_0$ gradient, and low-rankness to enhance detection accuracy and efficiency. In a recent study, the Hyperspectral Simultaneous Anomaly Detection and Denoising (HyADD) framework [21] integrates anomaly detection and denoising in a single framework, leveraging spatial-spectral gradient domain-based smoothness and subspace domain-based low-rankness to improve detection performance while removing additive noise. HyADD uses an adaptive dictionary construction to enhance the robustness of anomaly detection and noise removal. However, these approaches often require complex optimization processes that can be computationally intensive, particularly for high-dimensional data. Additionally, while they effectively model linear relationships, they may struggle to fully capture the nonlinear relationships that are

common in complex hyperspectral scenes. These limitations highlight the need for a method that can efficiently handle high-dimensional data and nonlinear relationships without compromising computational efficiency.

Deep learning methods have shown promise in hyperspectral anomaly detection [22,23]. Transferred deep convolutional neural networks (CNND) [24] classify pixel pairs using labeled reference data, accurately localizing anomalies. However, CNND relies heavily on surrounding pixels, making it susceptible to noise and local irregularities. Self-supervised networks such as the blind-block reconstruction network (BockNet) [25] and the pixel-shuffle downsampling blind-spot reconstruction network (PDBSNet) [26] address this issue by introducing blind-spot architectures to reduce the impact of anomalies on reconstruction results. The sliding dual-window-inspired reconstruction network (DirectNet) [27] further improves anomaly suppression by reconstructing central pixels using only outer window pixels. The nonlocal and local feature-coupled self-supervised network (NL2Net) [28] integrates local and nonlocal feature extraction to capture fine-grained spatial-spectral details and model long-range dependencies. Additionally, an improved center block masked convolution enhances the network's focus on surrounding background features, enabling precise background reconstruction and superior anomaly separation. More recently, the global feature-injected blind-spot network (PUNNet) [29] incorporates patch-shuffle downsampling and nonlinear activation-free network (NAFNet) blocks with dilated convolution to capture both local and global spatial-spectral features, achieving superior detection performance by reliably reconstructing the background while weakening the expression of anomalous features. Another innovative approach, the frequency-to-spectrum mapping generative adversarial network (FTSGAN) [30], maps the original spectra to the fractional Fourier domain to enhance the separability of backgrounds and anomalies, using a semisupervised learning strategy to prevent the model from focusing on numerical equivalence between input and output, thereby improving anomaly detection accuracy. Despite these advancements, existing deep learning methods often struggle to fully exploit the spatial-spectral relationships in hyperspectral images and may be computationally intensive, particularly for high-dimensional data. Moreover, while these methods have shown significant improvements in anomaly detection, they often rely on complex network architectures that may not be easily interpretable or adaptable to different types of hyperspectral data. There is a need for a method that can efficiently integrate spatial and spectral information, handle high-dimensional data, and provide a more interpretable and adaptable framework for anomaly detection.

Graph neural networks (GNNs) have recently been utilized to analyze hyperspectral images, predominantly for classification purposes rather than for anomaly detection [31–34]. Traditional methods use surrounding pixels to discriminate anomalous targets, which can be easily affected by anomalous targets and noise. Anomalous targets may appear at any position in the image and may even span the entire image. Furthermore, nonlinear relationships frequently occur in scenes characterized by intricate spatial distributions and numerous ground objects. GNNs proficiently address non-linear relationships by integrating both distant and adjacent pixels through the graph's topological structure, thereby enhancing the precision of anomaly detection. We propose a graph attention network (GAN) combined with a beta wavelet graph neural network (BWGNN) for hyperspectral anomaly detection, wherein each pixel in the hyperspectral image is regarded as a node, the relationships between pixels are depicted as edges, and the spectral characteristics of each pixel are represented as node attributes. By leveraging both spatial and spectral information, the algorithm provides a comprehensive analysis of the hyperspectral data. The contributions of this article are summarized as follows:

1. HSIs are graph represented with spatial and spectral information. GNNs use node - to - node learning to handle the nonlinear relationships in images, enhancing the detection of anomalous targets. The k - nearest neighbor algorithm is used to establish node edges by calculating pixel value distances, including distant pixels in detection and reducing the impact of noise and nearby anomalies on performance. The GAN–BWGNN HAD HAD algorithm treats each pixel as a graph node, leveraging both spatial and spectral information in HSIs for more accurate anomaly detection.

2. The GAN component in the algorithm features an adaptive attention mechanism. This mechanism dynamically weights the importance of neighboring nodes, enabling the algorithm to focus on relevant spatial features and improve sensitivity to spatially localized anomalies.

3. The BWGNN component creates a localized bandpass filter using beta wavelets to capture the right - shifted spectral energy phenomenon, distinguishing anomalous pixels from the background based on spectral signatures. Moreover, the proposed algorithm uses a beta wavelet as the spectral filter, avoiding the computational complexities of Laplace matrix decomposition and multiplication, thus enhancing hyperspectral data processing efficiency.

The remainder of this paper is organized as follows: The proposed algorithm section details the proposed GAN-BWGNN HAD methodology, including hyperspectral graph construction, spatial modeling via graph attention networks, and spectral anomaly detection using beta wavelet transforms. The Experiments section presents comprehensive experiments on seven benchmark datasets, evaluating detection performance, computational efficiency, parameter sensitivity, and ablation studies. Finally, the Conclusions section concludes the work and discusses future research directions.

## Proposed algorithm

### Graph construction of hyperspectral images

In this paper, the 3D hyperspectral data cube is set as $\widetilde{\mathcal{X}} \in \mathbb{R}^{n_1 \times n_2 \times d}$, $N = n_1 \times n_2$ is the number of pixels, $d$ is the number of spectral bands, and $X_i = [x_{i1}, x_{i2}, \dots, x_{id}] \in \mathbb{R}^d$ is the spectral vector of a pixel in $\widetilde{\mathcal{X}}$. The hyperspectral image is represented by the graph data structure, where the graph is defined as $\mathcal{G} = \{\mathcal{V}, \mathcal{E}, X\}$, taking each pixel as a node $v_i$, $\mathcal{V} = \{v_1, v_2, \cdots, v_N\}$ is a set of nodes, with the number of nodes being denoted as $N$, $\mathcal{E}$ is a set of edges, and $e_{ij} = (v_i, v_j) \in \mathcal{E}$ indicates that node $v_i$ and node $v_j$ are associated having an edge in between. The KNN algorithm is used to establish the edges between nodes. For each node, the pixel value disparity is computed between that node and other nodes. k-closest nodes are investigated, an edge is established among them, and the adjacency matrix $A$ is constructed. In most hyperspectral abnormal target detection algorithms, this is usually conducted using the surrounding pixels. However, noise and abnormal targets in surrounding pixels influence the detection performance. Using the KNN algorithm, the remote pixel can be involved in the detection, avoiding the interference of noise and abnormal targets in the neighborhood. $X = \{X_1, X_2, \cdots, X_N\}$ is the feature matrix, and the spectral vector $X_i$ of each pixel is used as a feature of the graph. The hyperspectral image is characterized by the graph data structure, which includes both the spatial information $\mathcal{E}$ and spectral information $X$ of the hyperspectral image. Considering $A$ to be the adjacency matrix and $D$, the degree matrix, one would have $D_{ii} = \sum_j A_{ij}$. The Laplacian matrix of the graph is defined as $L = D - A$, and symmetric normalized Laplacian is $L^{\mathrm{sys}} = D^{-1/2}LD^{-1/2} = I - D^{-1/2}AD^{-1/2}$, where $I$ is an identity matrix.

Eigenvalue decomposition for $L$ is performed as follows: where $0 = \lambda_1 \leq \cdots \leq \lambda_N$ are the eigenvalues, the corresponding eigenvector $U = (u_1, u_2, \cdots, u_N)$, $U \in \mathbb{R}^{n \times n}$ is an orthogonal matrix. Except for $\lambda_1$ and $\lambda_N$, the threshold $\lambda_k$ is arbitrarily selected to divide the eigenvalues into low frequencies $\{\lambda_1, \lambda_2, \cdots, \lambda_k\}$ and high frequencies $\{\lambda_{k+1}, \lambda_{k+2}, \cdots, \lambda_N\}$. Assuming that $x = (x_1, x_2, \cdots, x_N)^T \in \mathbb{R}^N$ is a signal on $\mathcal{G}$, the Fourier transform of $x$ is obtained as follows: $\hat{x} = (\hat{x}_1, \hat{x}_2, \cdots, \hat{x}_N)^T = U^T x$. The pseudocode of the algorithm is shown in Algorithm 1.

**Algorithm 1. Graph construction of hyperspectral images.**

**Require:**

1: 3D hyperspectral data cube $X \in \mathbb{R}^{n_1 \times n_2 \times d}$

2: Number of nearest neighbors k

**Ensure:**

3: Graph $G = \{V, E, X\}$, where V is the set of nodes, E is the set of edges, and X is the feature matrix.

4: Flatten the 3D data cube X into a 2D matrix $X \in \mathbb{R}^{N \times d}$, where $N = n_1 \times n_2$.

5: Treat each pixel as a node $v_i \in V$.

6: Use KNN algorithm to compute edges E between nodes based on pixel values.

7: Construct adjacency matrix A and degree matrix D.

8: Compute Laplacian matrix $L = D - A$.

9: **return** Graph $G = \{V, E, X\}$

## Graph attention network

GANs [35] leverage the attention mechanism to dynamically assign importance to the nodes' neighbors during feature aggregation, allowing a more nuanced representation of graph-structured data and adaptation to the specificities of each node's local neighborhood. The core idea behind GANs is to compute attention coefficients that reflect the importance of each neighbor's features to a node. These coefficients are calculated as follows:

Linear Transformation: First, a shared linear transformation, parameterized by a weight matrix $\mathbf{W} \in \mathbb{R}^{F' \times F}$, is applied to every node's features $h_i \in \mathbb{R}^F$, where $F'$ is the size of the new feature space, and $F$ is the size of the original feature space. This transformation is described using:

$$h_i' = \mathbf{W} h_i \tag{1}$$

This step maps the node features into a space where the attention mechanism can be applied more efficiently.

Pairwise Attention Scores: For each pair of nodes, an attention mechanism $a : \mathbb{R}^{F'} \times \mathbb{R}^{F'} \to \mathbb{R}$ computes a raw attention score $e_{ij}$ that indicates the importance of node $j$'s features to the node $i$, described as follows:

$$e_{ij} = a(\mathbf{W} h_i, \mathbf{W} h_j) \tag{2}$$

A common choice for the attention mechanism $a$ is a single-layer feed forward neural network, parameterized by a weight vector $\mathbf{a} \in \mathbb{R}^{2F'}$, and by applying the LeakyReLU nonlinearity:

$$e_{ij} = \text{LeakyReLU}\left(\mathbf{a}^T[\mathbf{W}h_i \| \mathbf{W}h_j]\right) \tag{3}$$

where $\|$ denotes concatenation.

Normalization of Attention Scores: The raw scores are then normalized across all choices of $j$ using the softmax function to facilitate straightforward comparison:

$$\alpha_{ij} = \frac{\exp(e_{ij})}{\sum_{k \in \mathcal{N}_i} \exp(e_{ik})} \tag{4}$$

where $\mathcal{N}_i$ denotes the set of neighbors of the node $i$ and $\alpha_{ij}$ represents the normalized attention coefficient that quantifies the importance of node $j$'s features to node $i$.

Feature Update: Finally, the node features are updated by computing a weighted combination of the neighbors' features using the normalized attention coefficients:

$$h_i' = \sigma\left(\sum_{j \in \mathcal{N}_i} \alpha_{ij} \mathbf{W} h_j\right) \tag{5}$$

where $\sigma$ denotes an activation function, and $h_i'$ is the updated feature vector of the node $i$.

Through these mechanisms, GANs adaptively focus on the most relevant parts of the graph structure for each node, leading to more effective learning from graph-structured data.

## Right-shift phenomenon

For any $1 \leq k \leq N-1$, the $k$-th low-frequency energy ratio as the accumulated energy distribution in the first $k$ eigenvalues, is described as follows [36]:

$$\eta_k(\boldsymbol{x}, \boldsymbol{L}) = \frac{\sum_{i=1}^k \hat{x}_i^2}{\sum_{i=1}^N \hat{x}_i^2} \tag{6}$$

A larger $\eta_k$ indicates that a larger part of the energy corresponds to the first $k$ eigenvalues. The calculation of the spectral energy rate depends on Laplacian matrix decomposition; however, matrix decomposition and multiplication increase computational complexity. To improve computational efficiency, a high-frequency area is introduced [36]. It is assumed that the low-frequency energy ratio curve $f(t)$ is defined as $f(t) = \eta_k(x, \boldsymbol{L})$, where $t \in [\lambda_k, \lambda_{k+1})$ and $1 \leq k \leq N-1$. The area between $f(t)$ and $g(t) = 1$ is defined as the high-frequency area:

$$S_{\text{high}} = \int_0^{\lambda_N} 1 - f(t)dt = \frac{\sum_{k=1}^N \lambda_k \hat{x}_k^2}{\sum_{k=1}^N \hat{x}_k^2} = \frac{\boldsymbol{x}^T \boldsymbol{L} \boldsymbol{x}}{\boldsymbol{x}^T \boldsymbol{x}} \tag{7}$$

The calculation of $S_{\text{high}}$ avoids Laplacian matrix decomposition and reduces computational complexity. In the work of [36], the authors proposed the idea of right-shift phenomenon. The presence of anomalies leads to a right shift in spectral energy, indicating that the spectral energy distribution is concentrated less at low frequencies and more at high frequencies. $S_{\text{high}}$ monotonically increases with the anomaly degree; therefore, $S_{\text{high}}$ can be used to represent the right-shift phenomenon and to measure the influence of the anomalous target in the spectral domain.

## GAN–BWGNN HAD

Because the graph spectral energy shows the right-shift phenomenon in the spectral domain, the graph signal $\boldsymbol{x}$ can be projected in the spectral domain via a Fourier transform. Nonetheless, the Fourier transform relies on the eigendecomposition of the Laplacian matrix, and the eigenvector is not a sparse matrix. Therefore, the calculation cost is exceedingly high regardless of whether eigendecomposition or eigenvector multiplication is performed. Moreover, the Fourier transform encompasses elements from the entire spectral domain, which does not possess adequate locality to accurately represent the right-shift phenomenon. The graph wavelet transform can overcome the limitations of the Fourier transform. This way, the Fourier transform basis can be replaced with the wavelet transform basis. A set of wavelets as the basis can be expressed as follows: $\mathcal{W} = \left(\mathcal{W}_{\psi_{s1}}, \mathcal{W}_{\psi_{s2}}, \cdots, \mathcal{W}_{\psi_{sN}}\right)$, where $\mathcal{W}_{\psi_{si}}$ is described as

$$\mathcal{W}_{\psi_{si}} = \boldsymbol{U}g_{si}(\Lambda)\boldsymbol{U}^T \tag{8}$$

where $\boldsymbol{U}$ is the eigenvector of the Laplacian matrix, $g_{si}(\Lambda)$ is a kernel function, $g_{si}(\Lambda) = \text{diag}\left(g\left(s\lambda_1\right), \ldots, g\left(s\lambda_N\right)\right)$, $s$ is a scale coefficient that describes different scales of the wavelet base, $g\left(s\lambda_i\right) = e^{\lambda_{si}}$. The wavelet transform of the graph signal $x$ based on $\mathcal{W}_{\psi_{si}}$ is described as follows:

$$\mathcal{W}_{\psi_{si}}(\boldsymbol{x}) = \boldsymbol{U}g_{si}(\Lambda)\boldsymbol{U}^T\boldsymbol{x}. \tag{9}$$

According to Parseval's theorem, the kernel function must satisfy the wavelet admissibility condition [37]:

$$C_g = \int_0^\infty \frac{\left|g_{si}(w)\right|^2}{w} dw < \infty \tag{10}$$

Here, $g_{si}(w)$ is a bandpass filter; $g_{si}(0) = g_{si}(\infty) = 0$. The wavelet function rapidly decays as it approaches 0 and infinity; therefore, the wavelet transform has good locality.

The graph wavelet transform projects the graph signal into the frequency domain via a Fourier transform and then filters the signal in the frequency domain via the wavelet function. Finally, the filtering result is projected back into the original domain via the Fourier transform. Owing to the right-shift phenomenon, the presence of anomalous targets leads to the concentration of spectral energy at high frequencies. Therefore, the selection of filters is important for capturing anomalous target signals. GNNs are usually low-pass filters or adaptive filters, which cannot capture the right-shift phenomenon. Furthermore, the kernel function selects a polynomial function or uses polynomial function approximation to avoid high computational complexity due to Laplacian matrix decomposition. Therefore, the beta distribution is selected as the graph wavelet base. The beta function is denoted by

$$B(a + 1, b + 1) = \int_0^1 w^a(1 - w)^b dw \tag{11}$$

The probability density function of beta distribution is described as follows:

$$\beta_{a,b}(w) = \begin{cases} \frac{1}{B(a+1,b+1)}w^{(a)}(1 - w)^{(b)} & \text{if } w \in [0, 1] \\ 0 & \text{otherwise} \end{cases} \tag{12}$$

where $a, b \in \mathbb{R}^+$, and $B(a + 1, b + 1) = \frac{a!b!}{(a+b+1)!}$ is a constant. As the eigenvalues of the normalized graph Laplacian $L$ satisfy $\lambda \in [0, 2]$, we adjust the probability density function of the beta

distribution as follows:

$$\beta_{a,b}^*(w) = \frac{1}{2}\beta_{a,b}\left(\frac{w}{2}\right) \tag{13}$$

where $a, b \in \mathbb{N}^+$, $\beta_{a,b}^*(w)$ is a polynomial, and Beta wavelet $\mathcal{W}_{a,b}$ can be written as

$$\mathcal{W}_{a,b} = \boldsymbol{U}\beta_{a,b}^*(\Lambda)\boldsymbol{U}^T = \beta_{a,b}^*(\boldsymbol{L}) = \frac{\left(\frac{L}{2}\right)^{(a)}\left(I - \frac{L}{2}\right)^{(b)}}{2B(a+1, b+1)} \tag{14}$$

Through recursive computation of the powers of $L$, such as $\left(\frac{L}{2}\right)^{(a)}$ and $(I - L/2)^{(b)}$, the polynomial kernel can be efficiently implemented. This approach avoids the high computational cost that would otherwise be incurred by explicit eigendecomposition. Let $a + b = C$ be a constant,the Beta wavelet transform $\mathcal{W}$ is constructed using a set of $C + 1$ Beta wavelets with the same order:

$$\mathcal{W} = (\mathcal{W}_{0,C}, \mathcal{W}_{1,C-1}, \cdots, \mathcal{W}_{C,0}) \tag{15}$$

In this equation, $\mathcal{W}_{0,C}$ is a low-pass filter, and the rest are bandpass filters of different scales. Besides, according to the Parseval theorem, when $a > 1$, the kernel function $\beta_{a,b}^*(w)$ satisfies the wavelet admissibility condition:

$$\int_0^\infty \frac{\left|\beta_{a,b}^*(w)\right|^2}{w}dw \leq \int_0^2 \frac{dw}{2B(a+1, b+1)} < \infty \tag{16}$$

Herein, we propose a novel HAD algorithm that combines the GAN and BWGNN to enhance detection performance. The algorithm works in two stages: spatial-domain processing and frequency-domain processing.

In the spatial-domain processing stage, the algorithm employs the GAN to capture the spatial information in the hyperspectral images. GANs utilize an attention mechanism to assign different weights to each pixel and its neighbors, effectively extracting local spatial features. This step enhances the model's understanding of the spatial context, laying the foundation for subsequent frequency-domain processing. The pseudocode of the algorithm is shown in Algorithm 2.

Thus, during the frequency-domain processing phase, the algorithm employs the BWGNN to examine the spectral attributes of hyperspectral images. Due to the right-shift phenomenon in spectral energy observed in hyperspectral data, the BWGNN employs the beta wavelet basis as a filter, which serves as an efficient and localized bandpass filter, particularly adept at capturing the right-shift in spectral energy. By processing in the frequency domain, the algorithm can more accurately identify and locate anomalous targets.

Overall, this combined approach of spatial and frequency domain processing provides a powerful tool for anomaly detection in hyperspectral images. It leverages the spatial information processing capabilities of the GAN and the frequency-domain analysis strengths of the BWGNN, enabling the algorithm to identify anomalies more effectively and thereby improving detection accuracy and efficiency. The network propagation process is described as follows [38]:

$$h_i = \boldsymbol{X}_i = [x_{i1}, x_{i2}, \ldots, x_{id}] \tag{17}$$

$$h_i' = \sigma\left(\sum_{j \in \mathcal{N}_i} \alpha_{ij}\boldsymbol{W}h_j\right) \tag{18}$$

**Algorithm 2. Spatial domain processing using graph attention network.**

**Require:**

1: Graph $G = \{V, E, X\}$

**Ensure:**

2: Updated node features $H' \in \mathbb{R}^{N \times d}$

3: Initialize node features $H = X$.

4: **for** each node $v_i \in V$ **do**

5:    Apply linear transformation: $h'_i = W \cdot h_i$, where W is a learnable weight matrix.

6:    Compute pairwise attention scores $e_{ij}$ for all neighbors $j \in N_i$:

7:    $e_{ij} = \text{LeakyReLU}(a^T[h'_i \| h'_j])$, where a is a weight vector.

8:    Normalize attention scores using softmax: $\alpha_{ij} = \frac{\exp(e_{ij})}{\sum_{k \in N_i} \exp(e_{ik})}$.

9:    Update node features: $h'_i = \sigma\left(\sum_{j \in N_i} \alpha_{ij} \cdot W h_j\right)$.

10: **end for**

11: **return** Updated node features $H'$

$$Z_i = \mathcal{W}_{i,C-i}(\text{MLP}(h'_i)) \tag{19}$$

$$S = \text{AGG}\left([Z_0, Z_1, \cdots, Z_C]\right) \tag{20}$$

$$P = \{p_1, p_2, \cdots, p_N\} = \text{MLP}(S) \tag{21}$$

$\mathcal{W}_{I,C-i}$ is a wavelet obtained from Eq. 15, $\text{MLP}(\cdot)$ is a multilayer perceptron, and $\text{AGG}(\cdot)$ is an aggregation function, such as summation or concatenation. The signal features enter the multilayer perceptron and are filtered using different wavelets in parallel. The filtering results are aggregated as $S$ and transmitted to the perceptron of the next layer. The sigmoid function is used as the network activation function. The network output is the anomaly probability $p_i$ of the pixels. The weighted cross-entropy loss is used for the training:

$$\mathcal{L} = \sum_i \left(\gamma y_i \log(p_i) + (1 - y_i) \log(1 - p_i)\right) \tag{22}$$

where $\gamma$ is the ratio of anomaly labels ($y_i = 1$) to normal labels ($y_i = 0$). The pseudocode of the algorithm is presented in Algorithm 3, and the flow chart of the proposed GAN–BWGNN HAD algorithm is shown in Fig 1.

## Experiments

The experiments were conducted on six real hyperspectral datasets and one simulated dataset. The hardware configuration consisted of an E5-2680v3 CPU, 128 GiB RAM, and an NVIDIA GeForce RTX 3060 GPU, with software implementation utilizing CUDA 11.6, PyTorch 1.12.0, and Python 3.7.

Hyperspectral images were modeled as homogeneous graphs based on pixel-level relationships. For the GABW-GNN HAD algorithm, model training was conducted using the Adam optimizer, with a learning rate of 0.01. The dimension of the network's hidden layer was set to 128, and the maximum number of epochs was set to 200. The training ratio was set to 60%. The order $C$ in the beta wavelet was set to 2, whereas $k$ in the KNN algorithm was set to 10. In GABW-GNN HAD, AGG $(\cdot)$ denotes concatenation. The evaluation framework compared the proposed method against three conventional detection algorithms (GRX [13], LRX [13],

**Algorithm 3. Frequency domain processing using beta wavelet graph neural network (BWGNN).**

**Require:**

1: Updated node features $H' \in \mathbb{R}^{N \times d}$

2: Beta wavelet parameters $a, b$

**Ensure:**

3: Anomaly probability map $P \in \mathbb{R}^N$

4: Define Beta wavelet basis $\beta_{a,b}^*(\Lambda)$ as per Equation (9).

5: **for** each node $v_i \in V$ **do**

6: Apply Beta wavelet filter to node features: $Z_i = \beta_{a,b}^*(L) \cdot$ MLP$(h_i')$.

7: Aggregate filtered features across all wavelet scales: $S = \text{AGG}([Z_0, Z_1, \dots, Z_C])$.

8: Compute anomaly probability: $p_i = \text{MLP}(S)$.

9: **end for**

10: **return** Anomaly probability map P

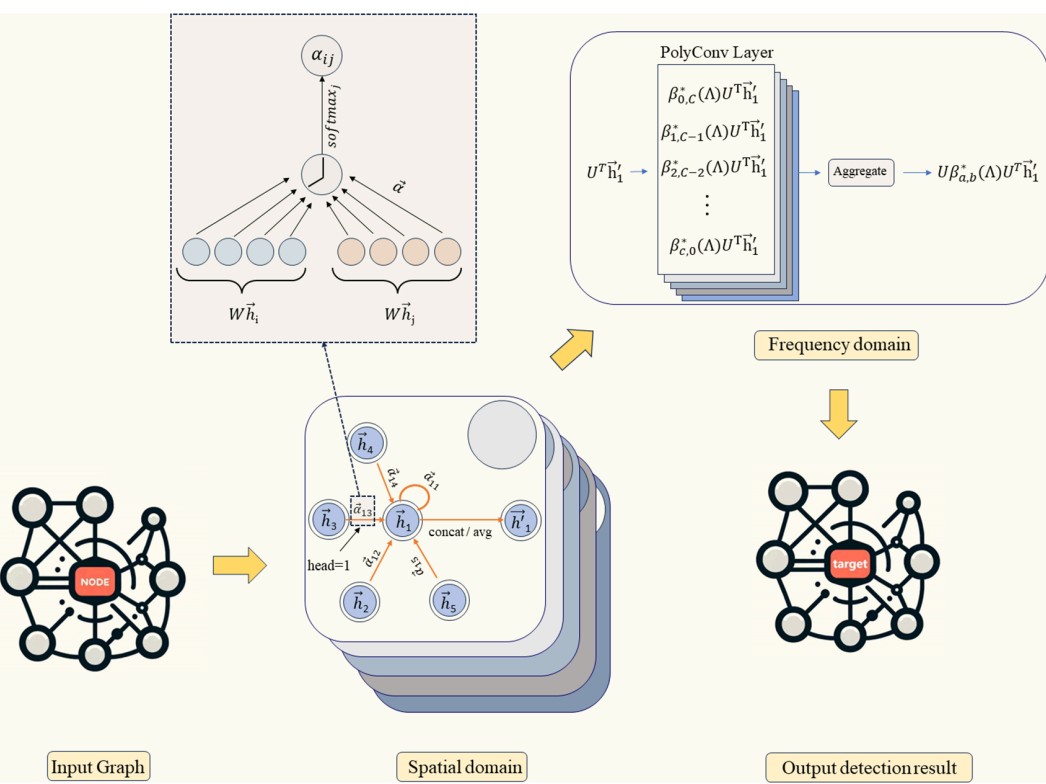

**Fig 1. Flowchart of the GAN–BWGNN HAD algorithm.**

and CRD [17]) and four advanced deep learning approaches (PDBSNet [26], BockNet [25], DirectNet [27], and NL2Net [28]). All comparative methods employed parameter settings consistent with their original publications to ensure fair benchmarking.

Quantitative assessment included AUC scores, 2D-ROC curves analyzing $(P_D, P_F)$, $(p_D, \tau)$, and $(p_F, \tau)$ relationships, 3D-ROC surface visualization, detection color maps for spatial

anomaly localization, background-anomaly separation boxplots, and runtime comparisons across all methodologies.

## Datasets

The datasets used in this study are publicly available and widely utilized in the field of hyperspectral anomaly detection. These datasets, including AVIRIS-I, AVIRIS-II, abu-beach-2, abu-urban-3, Cri, and GrandIsle, are real-world datasets that generally come with pre-labeled information. The labeled data are typically provided by the dataset creators based on known targets in real-world scenes (e.g., airplanes, vehicles, rocks) or through simulated embedding of anomalies. The Salinas-simulate dataset, which is a simulated dataset, has anomalies generated using a linear mixing model, and their locations are explicitly labeled.

1) AVIRIS-I: Captured by the NASA Airborne Visible/Infrared Imaging Spectrometer (AVIRIS) sensor, this dataset has a spatial resolution of 3.5 m per pixel and contains 224 spectral bands covering a wavelength range of 370 to 2510 nm.After eliminating bands affected by low signal-to-noise ratio, sensor issues, and water vapor absorption, 189 bands remained. A sub-image (120 × 120 pixels) was extracted from the upper-left corner of the original 400×400 pixel image. The dataset features diverse ground objects, including three airplanes, represented by 58 pixels as anomalies.

2) AVIRIS-II: This dataset was obtained from the San Diego Naval Airport, with a spatial resolution of 3.5 m per pixel. A 100×100 pixel sub-image was selected for the experiment. The scene contains various ground objects and a clear spectral distinction between the targets (three airplanes, occupying 134 pixels) and the background, with the target having a well-defined morphology.

3) abu-beach-2: Part of the Airport–Beach–Urban (ABU) dataset [39]. This scene was captured from the beach area in San Diego using the AVIRIS sensor. The dataset has a spatial resolution of 7.5 m per pixel and consists of 100×100 pixels. After removing noise-affected bands, 193 spectral bands remained. The anomalies in this scene were identified as fishing areas.

4) abu-urban-3: Another scene from the Airport-Beach-Urban dataset [39]. This urban scene was captured over Gainesville using the AVIRIS sensor. It has a resolution of 3.5 m per pixel and contains 191 spectral bands. The 100×100 pixel scene includes diverse land covers, such as buildings, roads, vegetation, and vehicles, with the vehicles being identified as anomalies.

5) Salinas-simulate: This dataset was created by embedding anomalous targets into the real Salinas dataset. Six square-shaped anomalous areas were arranged in reverse order, in close proximity to one another. The anomaly spectral signature was generated using a linear mixing model.

6) Cri: Captured by the Nuance Cri hyperspectral sensor [40], this large-scale dataset has a size of 400×400 pixels and includes 46 spectral bands from 650 to 1100 nm. Ten rock anomalies, represented by 2216 pixels, were identified in the scene.

7) Grand Isle: This dataset was captured over Grand Isle, Louisiana, USA, using NASA's Airborne Visible/Infrared Imaging Spectrometer (AVIRIS) sensor. It has a spatial resolution of 4.4 m per pixel and contains 224 spectral bands spanning the 366-2496 nm wavelength range.The scene (300×480 pixels) features islands, seawater, and artificial platforms in the Gulf of Mexico. The anomalies comprise artificial structures, predominantly oil/gas platforms situated offshore.

## Detection results

In the following Table 1, the AUC values of the GABW-GNN HAD are compared with those of eight other methods.

The comparative experimental results indicated that for the AVIRIS-I dataset, the LRX and CRD algorithms achieved optimal results with dual window settings of (31, 13) and (17, 15), respectively. As evident from Table 1, the CRD algorithm performs the best, achieving an AUC value of 0.9968. In comparison, the proposed algorithm achieves the second-best performance with an AUC value of 0.9937, showing only a marginal difference of 0.0031 from the CRD algorithm. The slightly lower AUC of the proposed method on the AVIRIS-I dataset compared to CRD can be attributed to the inherent characteristics of the dataset and algorithmic trade-offs. The AVIRIS-I scene contains three compact anomalies (aircrafts, 58 pixels) with high spectral contrast against a homogeneous background. CRD, a local linear collaborative representation method, excels in such scenarios by leveraging the strong correlation between anomalies and their immediate spatial neighbors. In contrast, proposed graph-based approach constructs edges via KNN, which introduces non-local pixel associations. While this global connectivity enhances resilience to noise and irregular backgrounds, it may dilute the localized discriminative features critical for detecting spatially compact anomalies. This suggests that the algorithm's strength in capturing global spatial-spectral dependencies may trade off sensitivity to highly localized targets. As shown in Fig 2, the proposed algorithm outperforms other methods in the 2D-ROC curve of $(P_D, P_F)$. Its curve lies above the others across most $P_F$ values, indicating higher detection probability ($P_D$) at lower false alarm rates ($P_F$). The CRD algorithm performs the second best, while the other algorithms show relatively inferior performance. Based on the PR curve, the proposed method exhibits sub-optimal yet robust performance: while slightly underperforming CRD (light pink curve) at specific points (e.g., a $P_D$ gap of 0.15 at $P_F = 10^{-3}$), it significantly outperforms other baseline methods across the $P_F$ range, demonstrating superior reliability in balancing detection accuracy and false alarm suppression. As shown in Fig 2, which presents the 2D-ROC $(p_D, \tau)$ curve, the curves of PDBSNet, BockNet, DirectNet, and NL2Net are overlapping, indicating similar detection performance. The proposed algorithm demonstrates outstanding performance in balancing detection probability and threshold robustness, particularly exhibiting competitive advantages in low to medium threshold ranges. The 2D-ROC $(p_F, \tau)$ curve in Fig 2 demonstrates that the proposed method maintains a significantly lower false positive rate ($p_F$) across the entire threshold range ($\tau$) compared to other methods, indicating its advantage in false alarm control. In Fig 3, the detection results for various algorithms are visualized via color maps. Both the GRX and LRX algorithms failed to detect the anomalies in the image. The DirectNet algorithm succeeded in pinpointing the locations of three airplanes; however, their shapes were not clearly defined. The NL2Net algorithm successfully detected the shapes and positions of the three airplanes, but it did not effectively suppress the background. CRD, PDBSNet, and

**Table 1. Comparison of AUC values.**

| Dataset | GRX | LRX | CRD | PDBSNet | BockNet | DirectNet | NL2Net | Proposed |
|---|---|---|---|---|---|---|---|---|
| AVIRIS-I | 0.9111 | 0.9605(31,13) | **0.9968**(17,15) | 0.9835 | 0.9887 | 0.9646 | 0.4453 | 0.9937 |
| AVIRIS-II | 0.9403 | 0.9336((35,13) | 0.9767(21,19) | 0.9820 | 0.9849 | 0.9713 | 0.7368 | **0.9986** |
| abu-beach-2 | 0.9106 | 0.9854(5,3) | 0.9680(7,3) | 0.9282 | 0.9794 | 0.9331 | 0.9037 | **0.9961** |
| abu-urban-3 | 0.9513 | 0.9337(35,33) | 0.9662(9,3) | 0.9647 | 0.9775 | 0.9686 | 0.7835 | **0.9982** |
| Salinas-simulate | 0.9861 | 0.9676(25,23) | 0.9821(15,3) | 0.9990 | 0.9993 | 0.9945 | 0.9501 | **0.9999** |
| GrandIsle | 0.9963 | 0.9936(11,9) | 0.9933(11,9) | 0.9599 | **0.9989** | 0.9614 | 0.8977 | 0.9966 |
| Cri | 0.9134 | 0.8579(25,21) | 0.9616(17,13) | 0.9566 | 0.8329 | 0.6176 | 0.9028 | **0.9872** |

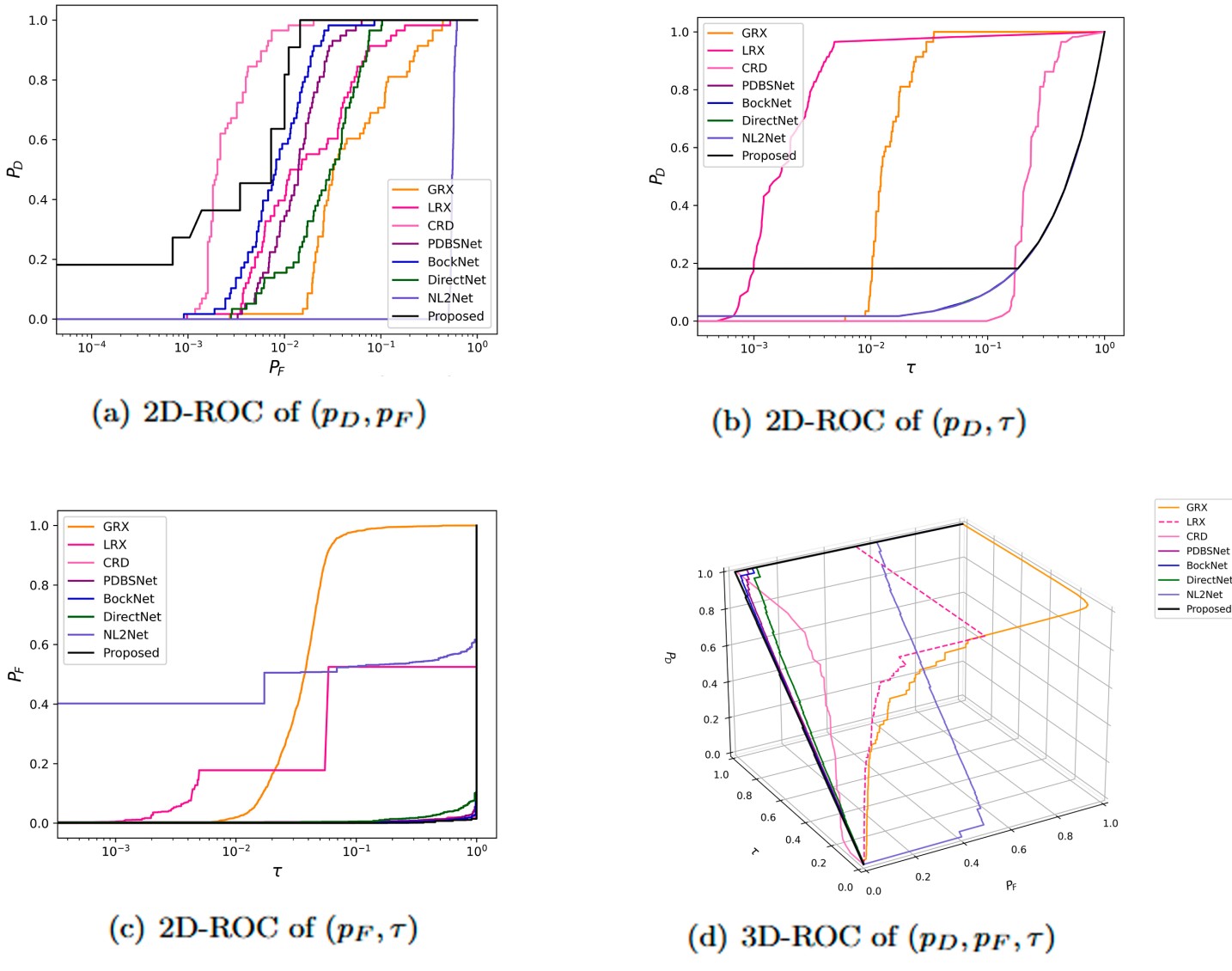

**Fig 2. Detection accuracy evaluation for the AVIRIS-I dataset.**

BockNet accurately identified the shapes and locations of the three airplanes. Nevertheless, they lacked sufficient prominence to distinguish themselves from the background, primarily because the algorithms failed to sufficiently enhance the contrast between the anomalies and the surrounding environment. The proposed algorithm generated distinct and vivid representations of the three airplanes, resulting in the highest accuracy in identification compared to the other techniques. Nonetheless, it inaccurately categorized a limited quantity of background pixels as anomalies. As shown in the boxplot (Fig 4), the proposed algorithm exhibits the largest separation between background and anomaly targets (maximum mean difference) compared to other methods, with both data distributions being highly compact (narrow box ranges and minimal outliers).

Experimental results demonstrated that for the AVIRIS-II dataset, the LRX and CRD algorithms achieved optimal performance with dual window settings of (35, 13) and (21, 19),

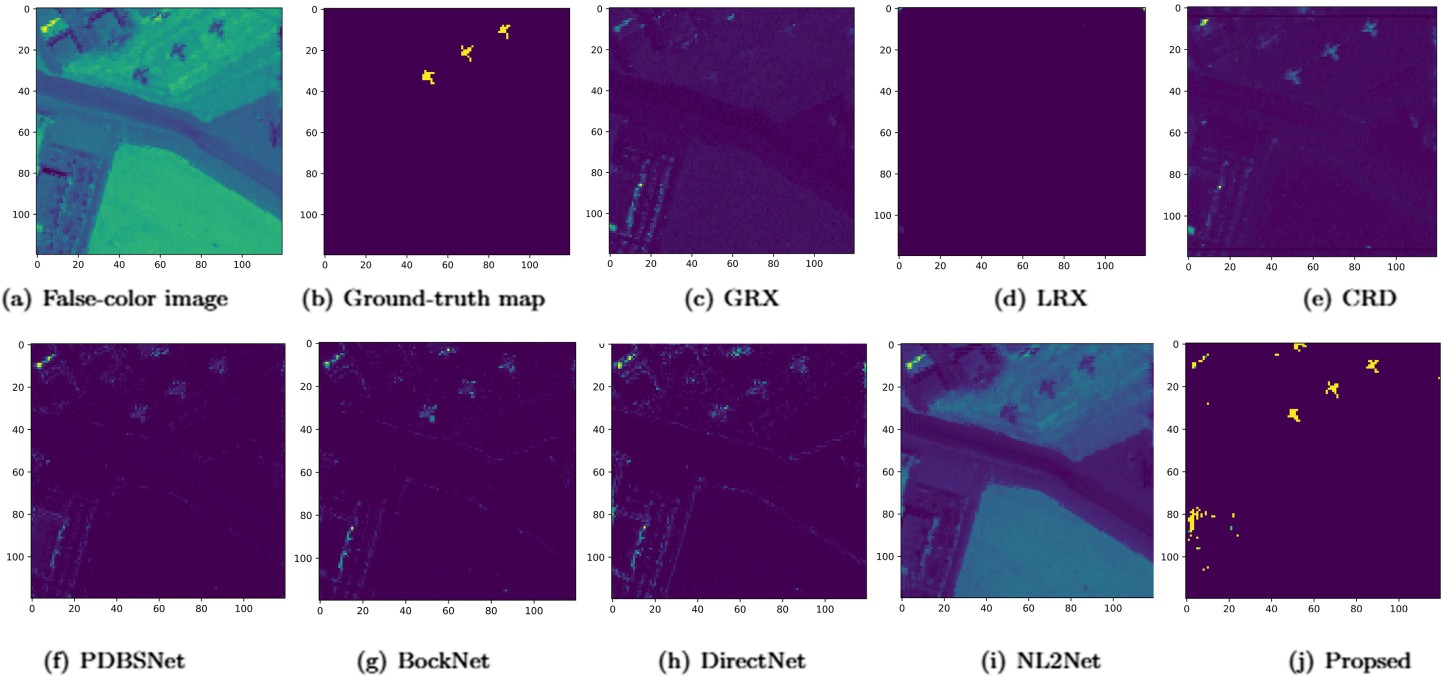

(a) False-color image
(b) Ground-truth map
(c) GRX
(d) LRX
(e) CRD

(f) PDBSNet
(g) BockNet
(h) DirectNet
(i) NL2Net
(j) Propsed

**Fig 3. Detection maps obtained using different algorithms for the AVIRIS-I dataset.**

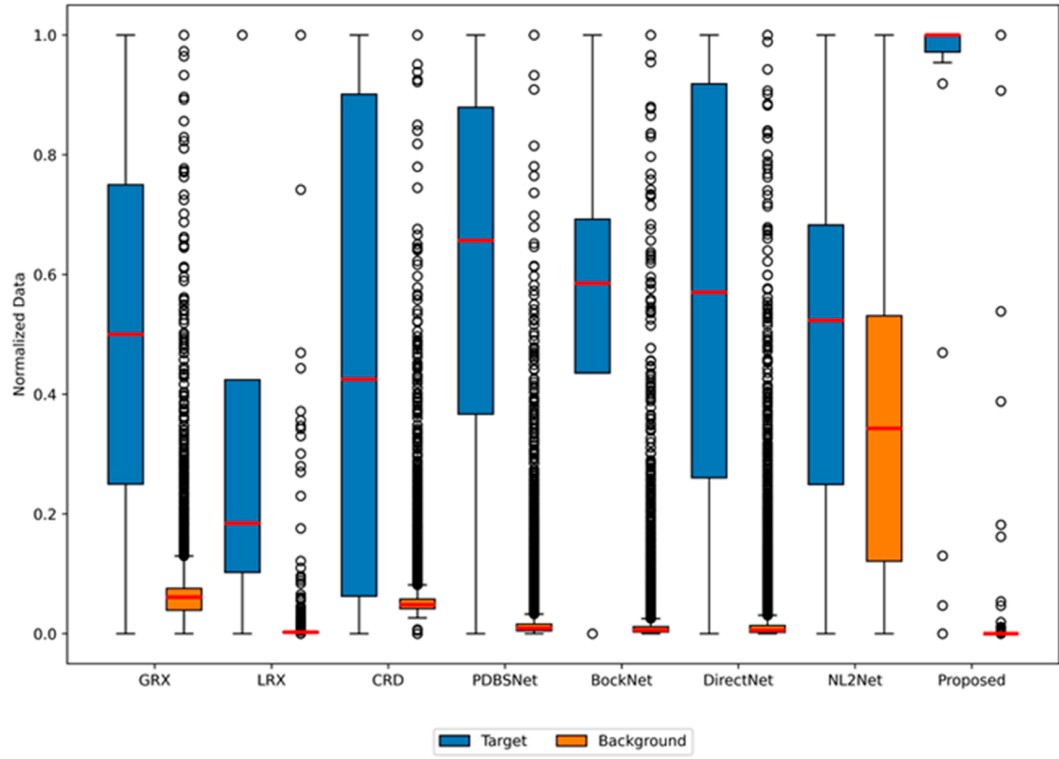

**Fig 4. Background-anomaly separation in AVIRIS-I dataset.**

respectively. As shown in Table 1, the proposed method exhibited the highest AUC value, surpassing all competing algorithms. Fig 5 presents 2D and 3D ROC curves comparing the performance of different methods for hyperspectral anomaly detection. In the 2D ROC curve $(P_D,P_F)$, the proposed method's curve was closest to the top-left corner, indicating the highest detection rate at equivalent false alarm rates, thereby outperforming other methods. In the 2D ROC curve $(P_D,\tau)$, the proposed method demonstrated superior performance under stringent conditions, with a rapid increase in detection rate at high thresholds ($\tau \geq 10^{-1}$). Additionally, in the 2D ROC curve $(P_F, \tau)$, the proposed method maintained the lowest false alarm rate. The 3D ROC curve further highlighted the proposed method's comprehensive advantages in balancing detection rate, false alarm rate, and threshold adaptability. Overall, the proposed method demonstrated the best performance in terms of detection rate, false alarm rate, and threshold optimization. The visualization of detection results using color maps in Fig 6 demonstrated the performance differences among various algorithms. The LRX algorithm failed to identify any anomalies in the scene. While GRX, CRD, PDBSNet, BockNet, and DirectNet were able to detect the locations of the three airplanes, they did not fully outline their shapes. NL2Net exhibited limited background suppression. In contrast, the proposed algorithm achieved the highest detection accuracy, accurately identifying the positions and shapes of the three airplanes while clearly distinguishing them from the background. Despite the presence of background noise, the proposed method demonstrated superior performance in hyperspectral anomaly detection compared to other algorithms. As shown in the boxplot (Fig 7), the proposed algorithm exhibits the largest gap between background and anomaly targets compared to other algorithms.

Experimental results demonstrated that for the abu-beach-2 dataset, the LRX and CRD algorithms achieved optimal performance with dual window settings of (5, 3) and (7, 3), respectively. As shown in Table 1, the proposed algorithm achieved the highest AUC value, indicating its superior performance over comparative methods. Fig 8 demonstrated the performance comparison of different methods for hyperspectral anomaly detection using 2D and 3D ROC curves. In the 2D ROC curve $(P_D,P_F)$, the proposed method achieved the highest detection rate at equivalent false alarm rates, as its curve was closest to the top-left corner, outperforming other methods. In the 2D ROC curve $(P_D,\tau)$, the proposed method exhibited superior performance under strict conditions, with a rapid increase in detection rate at high thresholds ($\tau \geq 10^{-1}$). Additionally, in the 2D ROC curve $(P_F,\tau)$, the proposed method maintained the lowest false alarm rate, indicating strong background suppression. The 3D ROC curve further highlighted the proposed method's comprehensive advantages in balancing detection rate, false alarm rate, and threshold adaptability. Overall, the proposed method demonstrated the best performance in terms of detection rate, false alarm rate, and threshold optimization. Fig 9 presents the detection color maps for various algorithms. The GRX algorithm successfully detected all anomalies but exhibited limited background suppression. PDBSNet, DirectNet, and NL2Net failed to identify all anomalies. While LRX, CRD, and BockNet were able to detect the positions of anomalies, the detected targets lacked clarity. The proposed method distinctly differentiated the anomalous objects from the background with remarkable brightness and clarity, despite a minor number of background pixels being erroneously labeled as anomalies. Fig 10 presents box plots comparing the separation of background and anomaly targets across different algorithms. The results show that NL2Net and PDBSNet achieved the highest separation, followed closely by the proposed method.

Experimental results demonstrated that for the abu-urban-3 dataset, the LRX and CRD algorithms achieved optimal performance with dual window settings of (35, 33) and (9, 3), respectively. As demonstrated in Table 1, the proposed algorithm achieved the highest AUC value, highlighting its superior detection capability. Fig 11 demonstrated the performance

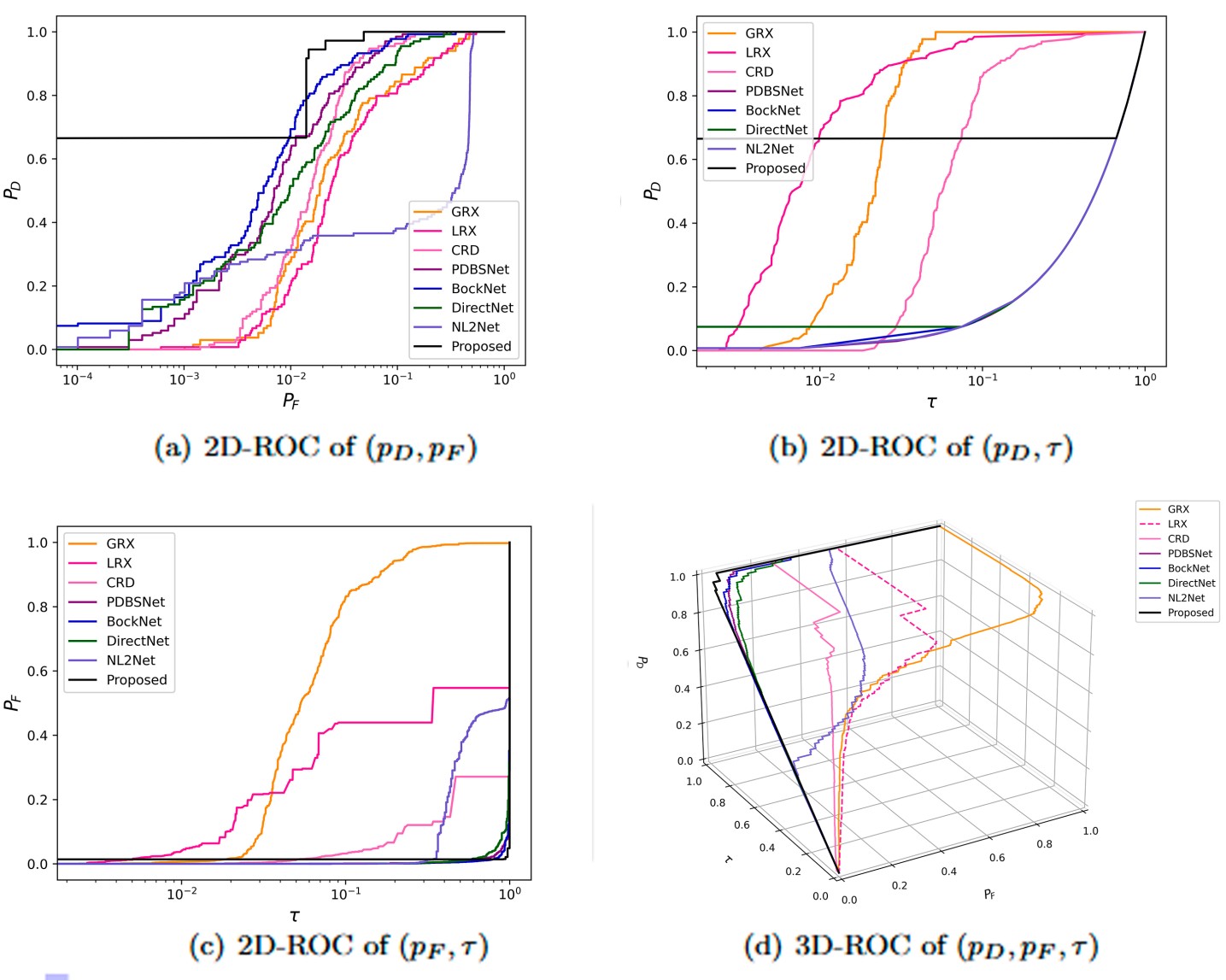

**Fig 5. Detection accuracy evaluation for the AVIRIS-II dataset.**

comparison of different methods for hyperspectral anomaly detection using 2D and 3D ROC curves. In the 2D ROC curve $(P_D, P_F)$, the proposed method's curve is closest to the top-left corner, indicating superior detection performance compared to other methods. In the 2D ROC curve $(P_D, \tau)$, the proposed method achieves a rapid increase in detection rate at high thresholds $(\tau \geq 10^{-1})$, demonstrating strong performance under strict conditions. In the 2D ROC curve $(P_F, \tau)$, the proposed method maintains the lowest false alarm rate across varying thresholds. The 3D ROC curve further highlights the proposed method's comprehensive advantages in balancing detection rate, false alarm rate, and threshold adaptability. Overall, the proposed method outperforms other algorithms in terms of detection rate, false alarm rate, and threshold optimization. The anomaly detection results visualized via color maps in

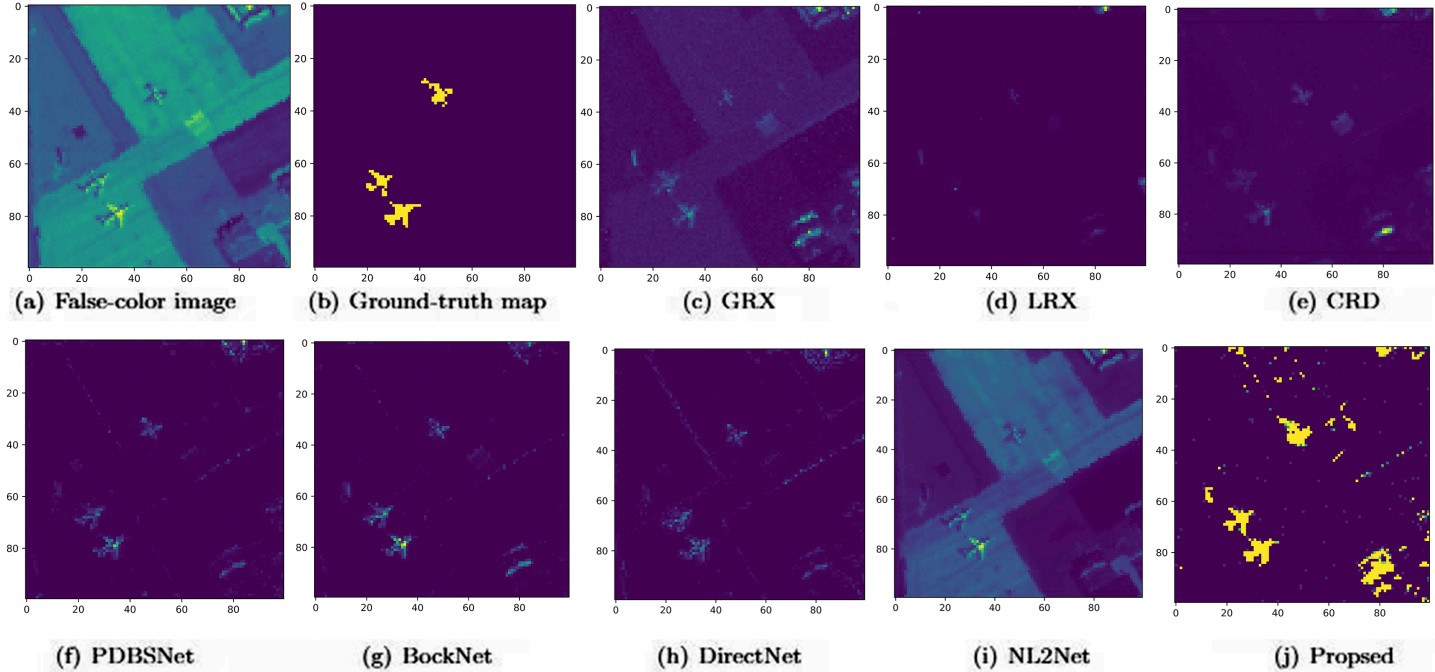

**Fig 6. Detection maps obtained using different algorithms for the AVIRIS-II dataset.**

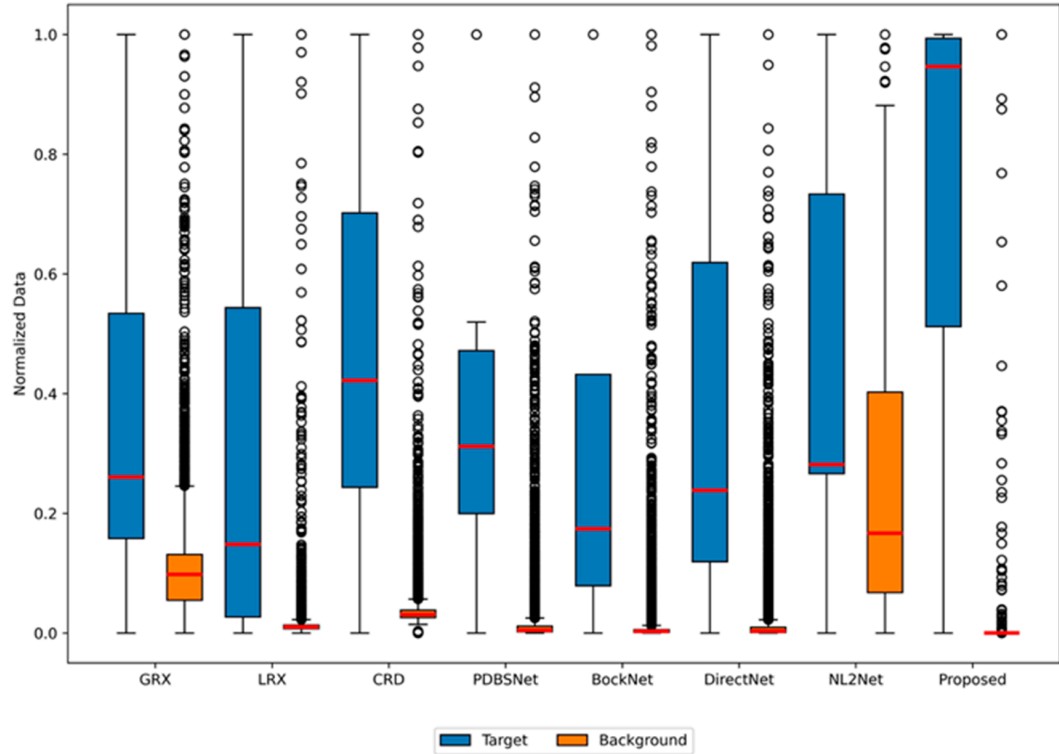

**Fig 7. Background-anomaly separation in AVIRIS-II dataset.**

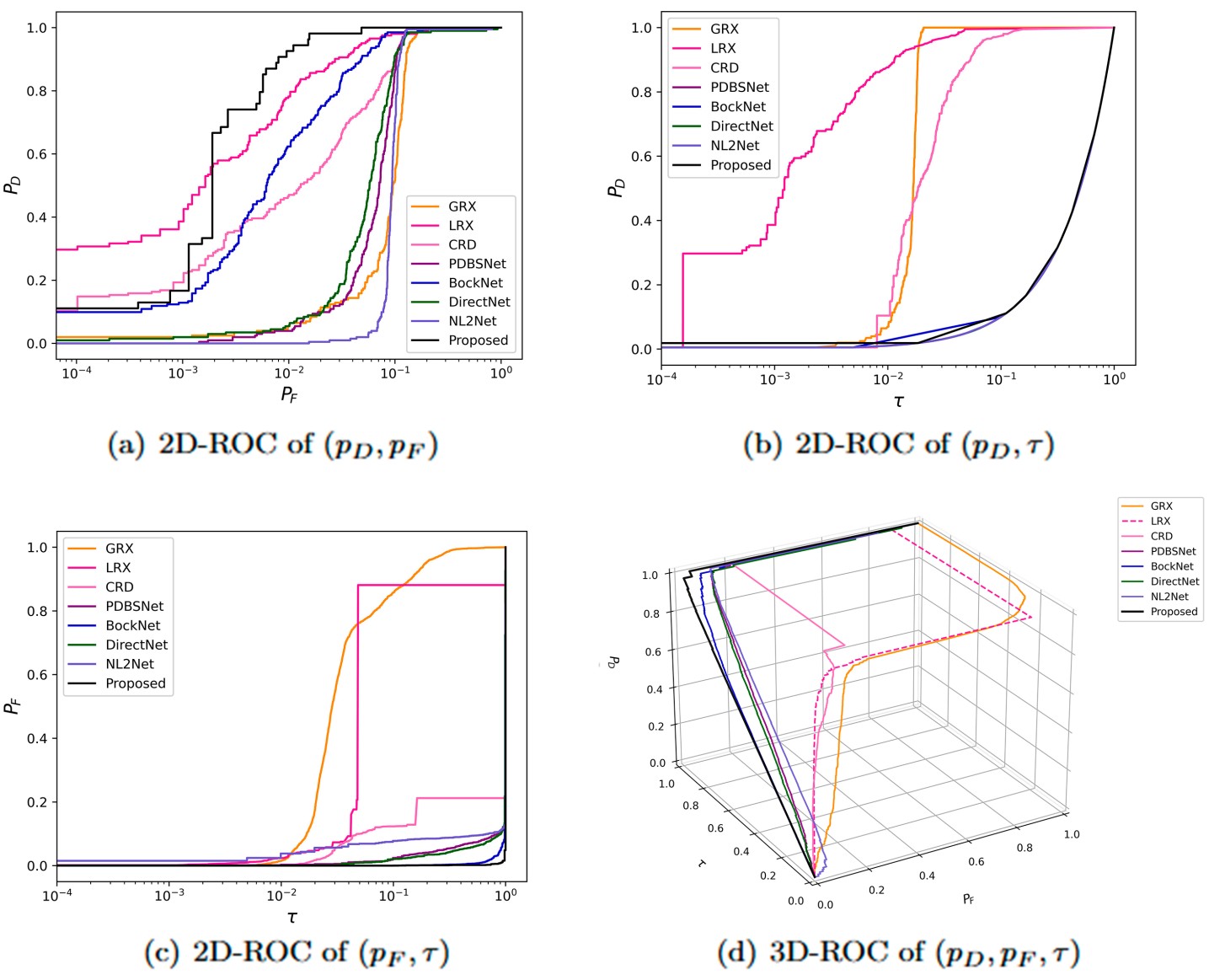

**Fig 8. Detection accuracy evaluation for the abu-beach-2 dataset.**

Fig 12 showed that the LRX algorithm failed to identify any anomalies within the scene. Algorithms such as GRX, CRD, PDBSNet, and NL2Net exhibited limited background suppression, leaving residual noise in the results. BockNet and DirectNet were able to detect anomalies but lacked clarity in outlining their shapes. In contrast, the proposed algorithm accurately identified the positions and shapes of all anomalies with higher clarity, though it had minor misclassifications. As shown in the boxplot (Fig 13), the proposed algorithm exhibits the largest gap between background and anomaly targets compared to other algorithms.

In experiments on the Salinas-simulate dataset, the optimal dual-window configurations for LRX and CRD were determined as (25, 23) and (15, 3), respectively. As shown in Table 1, the proposed algorithm achieved the highest AUC value, surpassing all baseline methods and

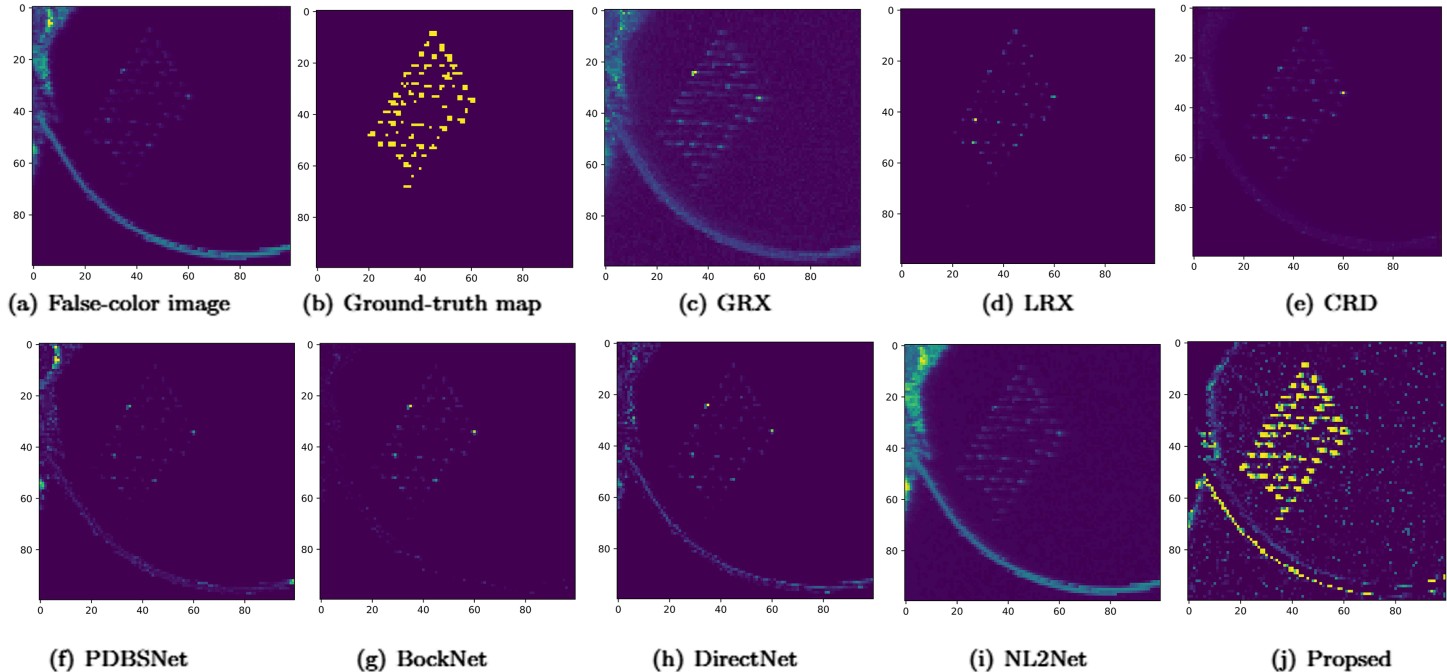

**Fig 9. Detection maps obtained using different algorithms for the abu-beach-2 dataset.**

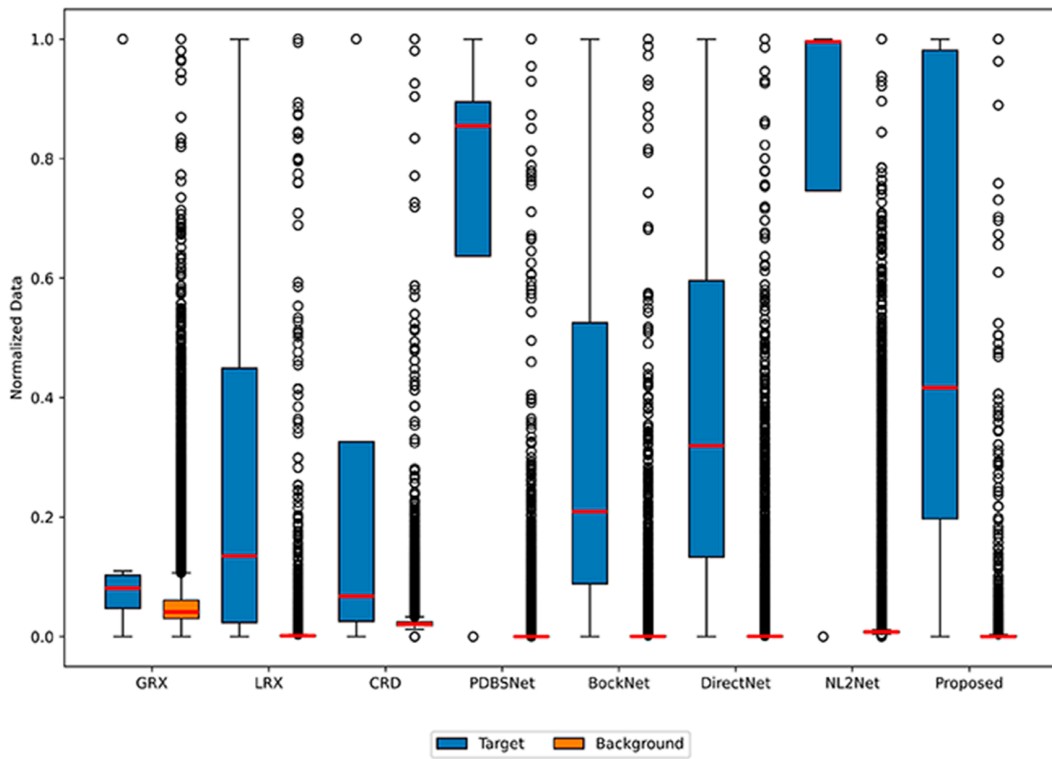

**Fig 10. Background-anomaly separation in abu-beach-2 dataset.**

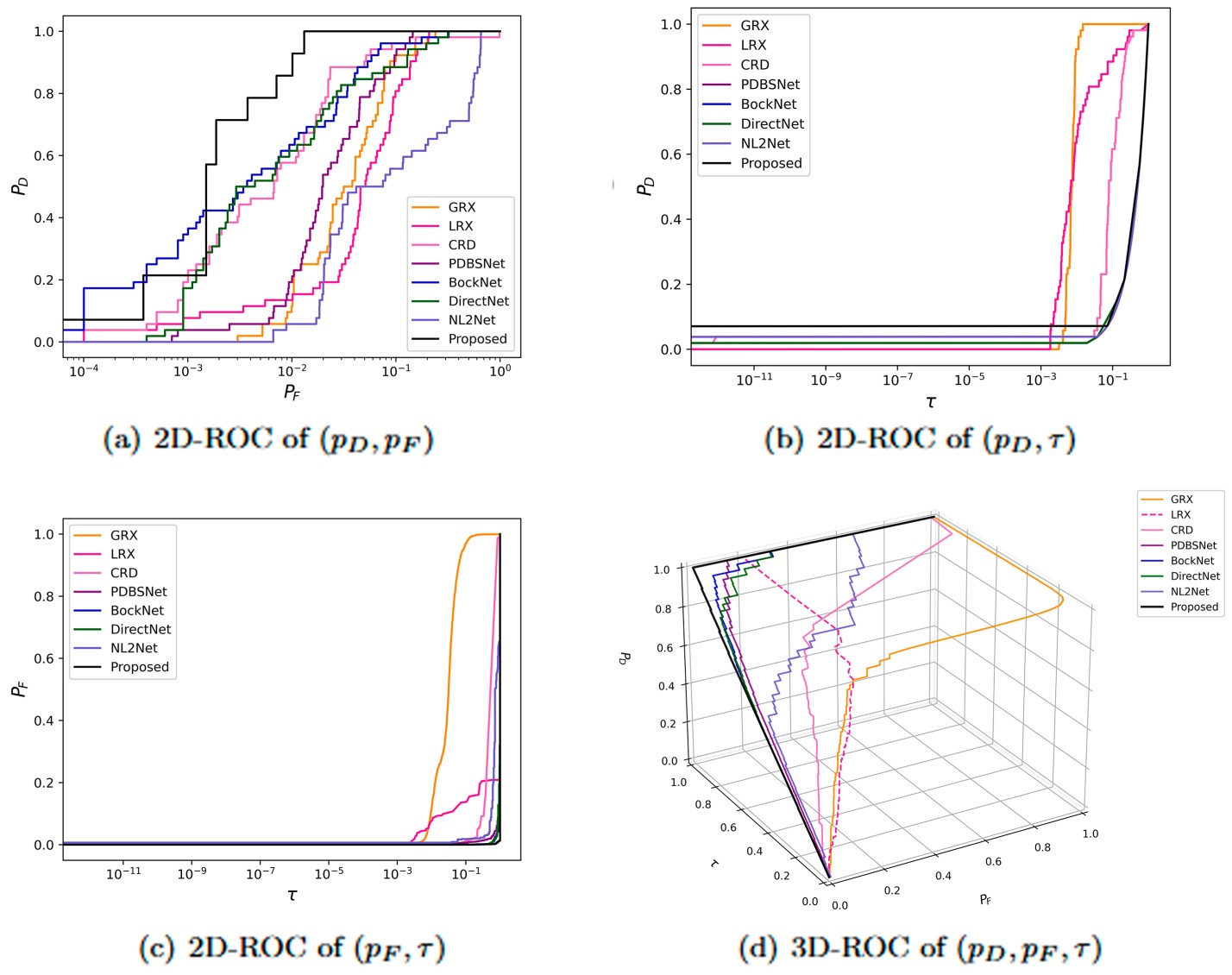

**Fig 11. Detection accuracy evaluation for the abu-urban-3 dataset.**

validating its enhanced detection accuracy. Fig 14 demonstrated the performance comparison of different methods for hyperspectral anomaly detection using 2D and 3D ROC curves. In the 2D ROC curve ($P_D$,$P_F$), the proposed method appears as a straight line consistently positioned at the top ($P_D \approx 1$), demonstrating its ability to achieve near-perfect detection performance across all $P_F$ ranges and significantly outperforming the other comparative methods. In the 2D ROC curve ($P_D$,$\tau$), the proposed algorithm forms a near-straight line at the top ($P_D \approx 1$) across the logarithmic threshold range ($\tau = 10^{-8}$ to $10^{0}$), demonstrating its superior and stable detection probability under compared to other methods. Additionally, in the 2D ROC curve ($P_F$,$\tau$), the proposed method maintains the lowest false alarm rate across varying thresholds, underscoring its robustness. The 3D ROC curve provides a comprehensive view, further emphasizing the proposed method's balanced performance in detection rate, false alarm rate, and threshold adaptability. Fig 15 displays the anomaly detection results on the

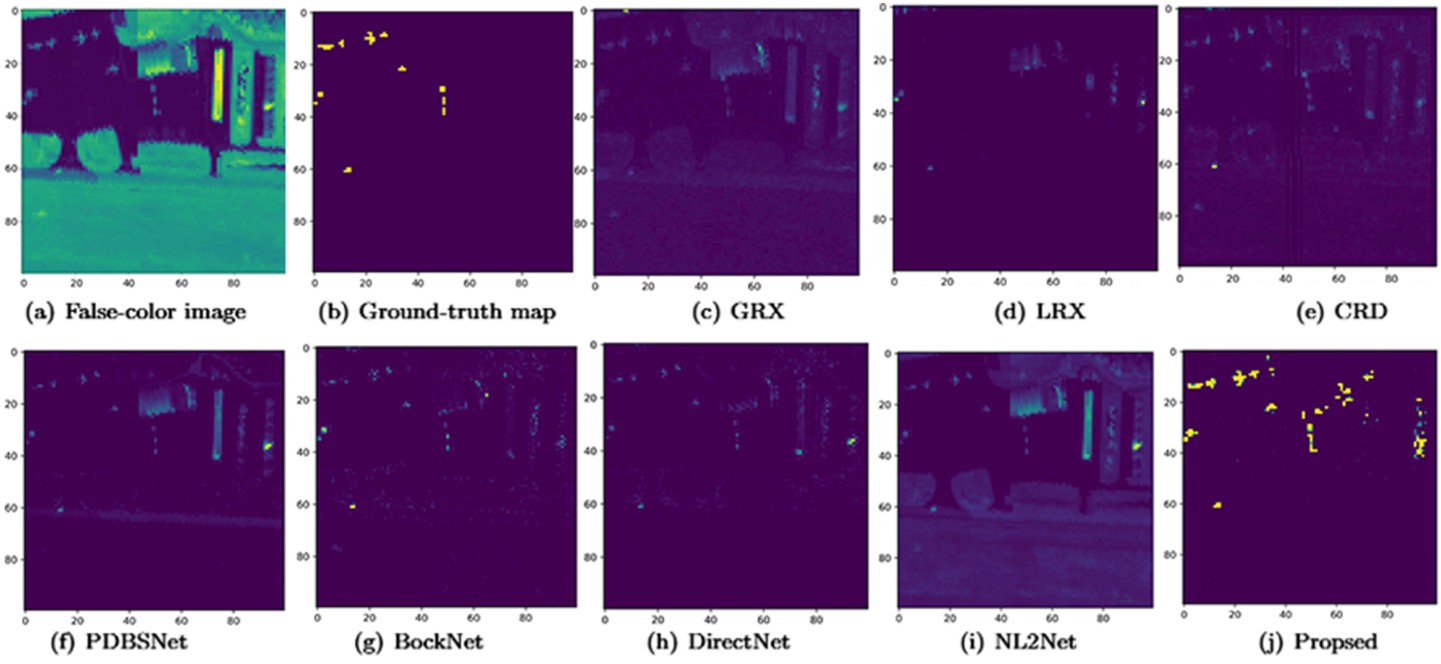

**Fig 12. Detection maps obtained using different algorithms for the abu-urban-3 dataset.**

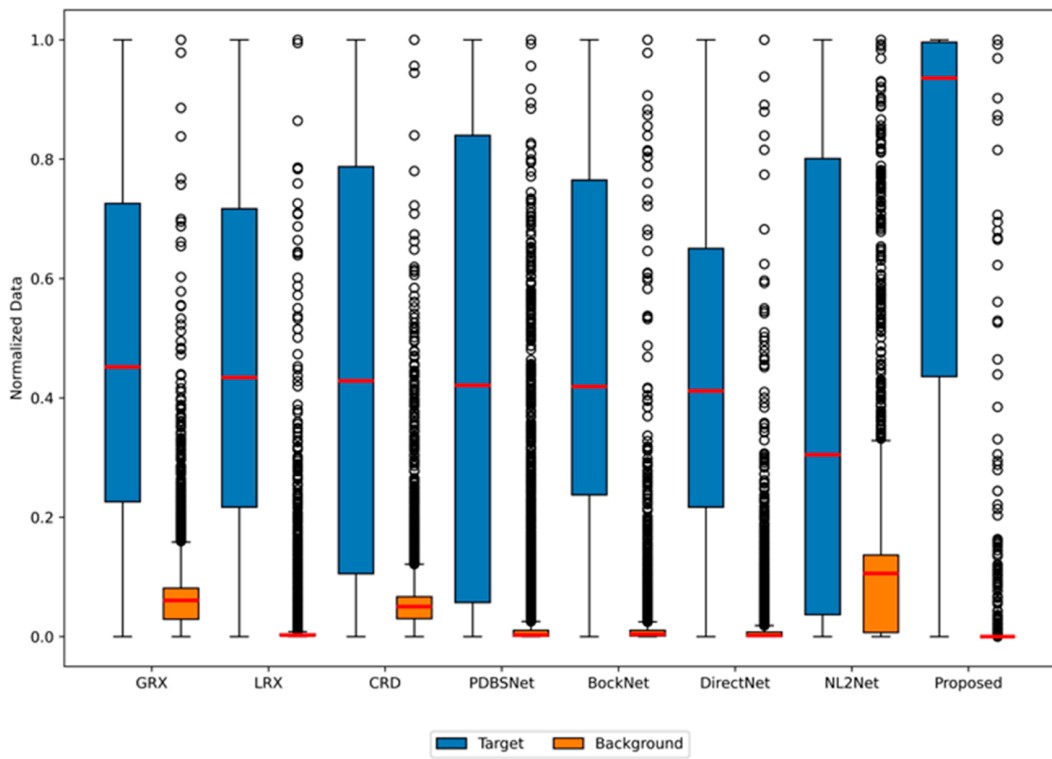

**Fig 13. Background-anomaly separation in abu-urban-3 dataset.**

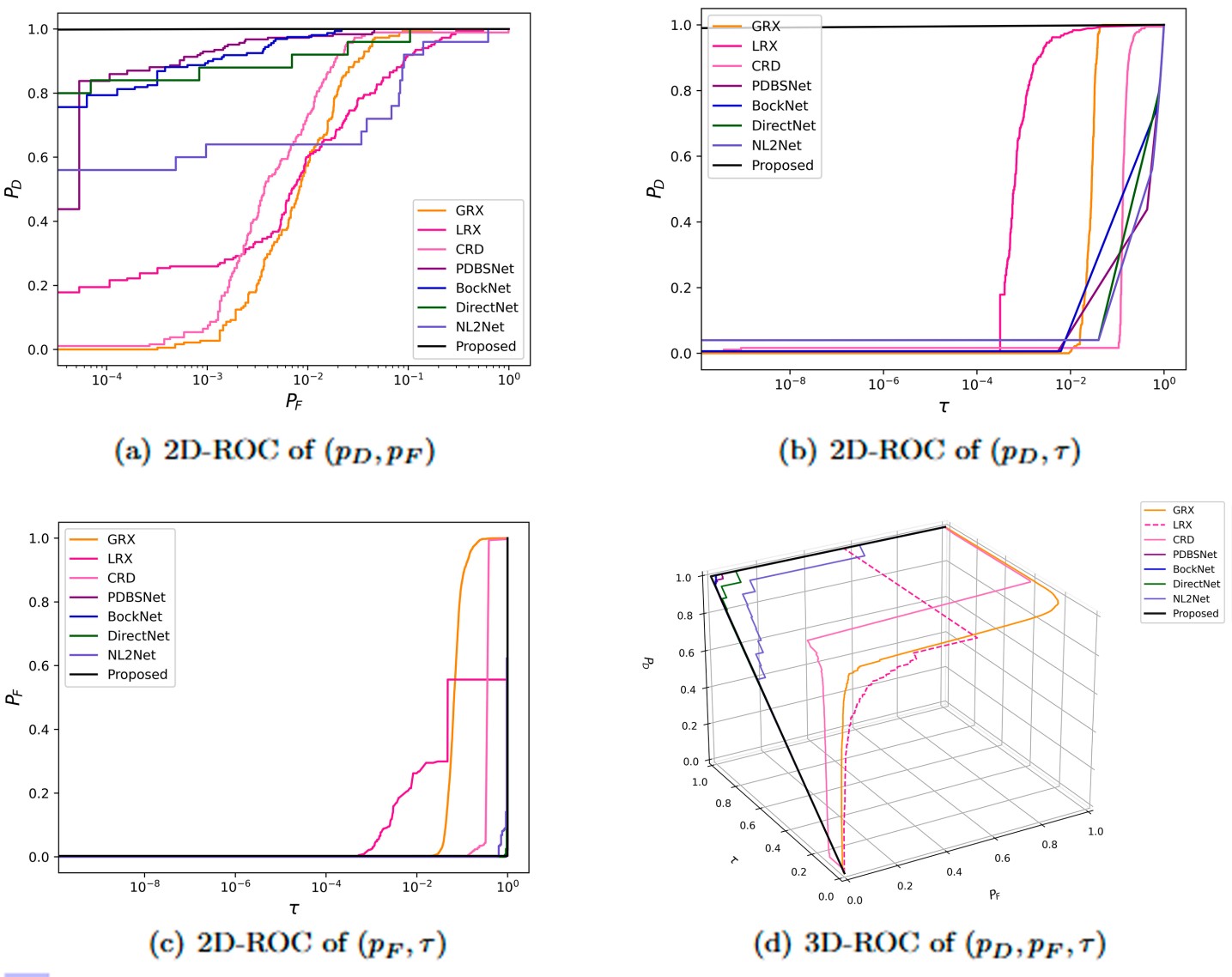

**Fig 14. Detection accuracy evaluation for the Salinas-simulate dataset.**

Salinas-simulate dataset. While LRX, CRD, and DirectNet failed to detect anomalies entirely, GRX and NL2Net exhibited limited separability between anomalies and background regions. PDBSNet and BockNet successfully identified all anomalous targets, though with partial background interference. In contrast, the proposed algorithm precisely localized both the positions and geometric contours of anomalies, effectively enhancing their visibility against the background with minimal interference. As shown in the boxplot (Fig 16), the proposed algorithm exhibits the largest separation between background and anomaly targets compared to other methods, with both data distributions being highly compact.

In experiments on the GrandIsle dataset, the optimal dual-window configurations for LRX and CRD were both determined as (11, 9). As shown in Table 1, The proposed method achieved an AUC of 0.9966, closely approaching BockNet's peak performance (0.9989) with

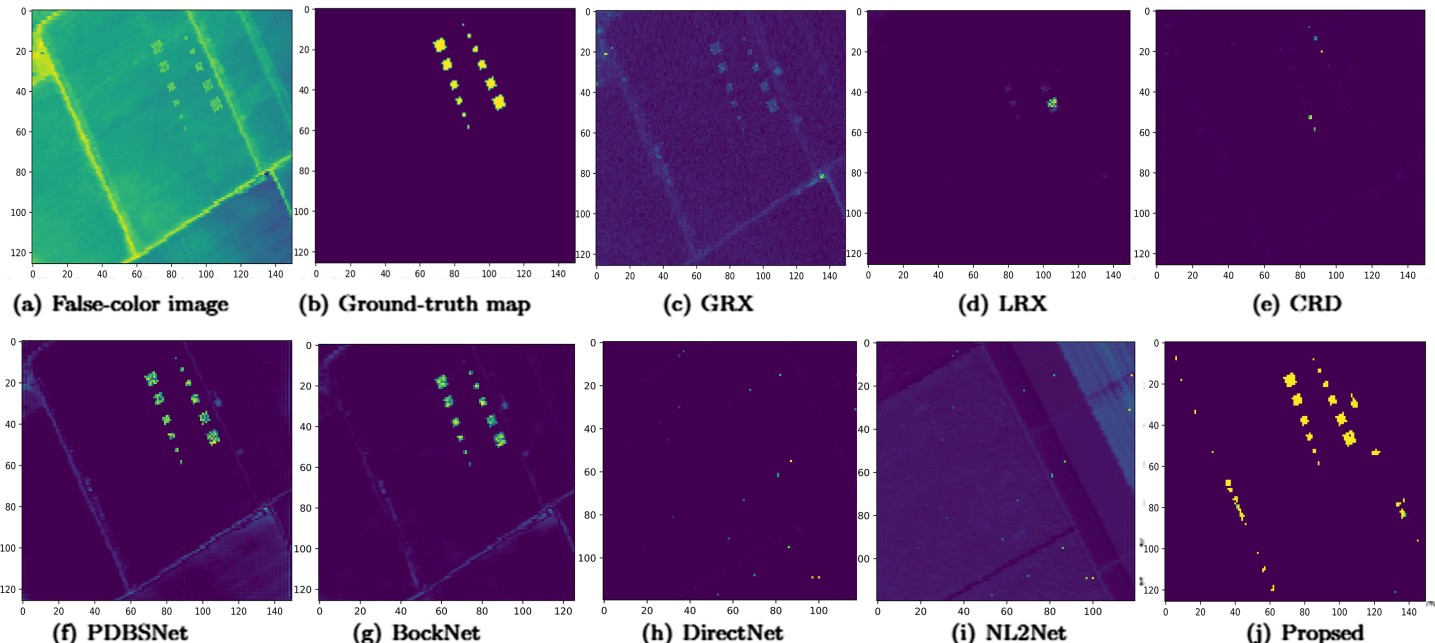

**Fig 15. Detection maps obtained by different algorithms for the Salinas-simulate dataset.**

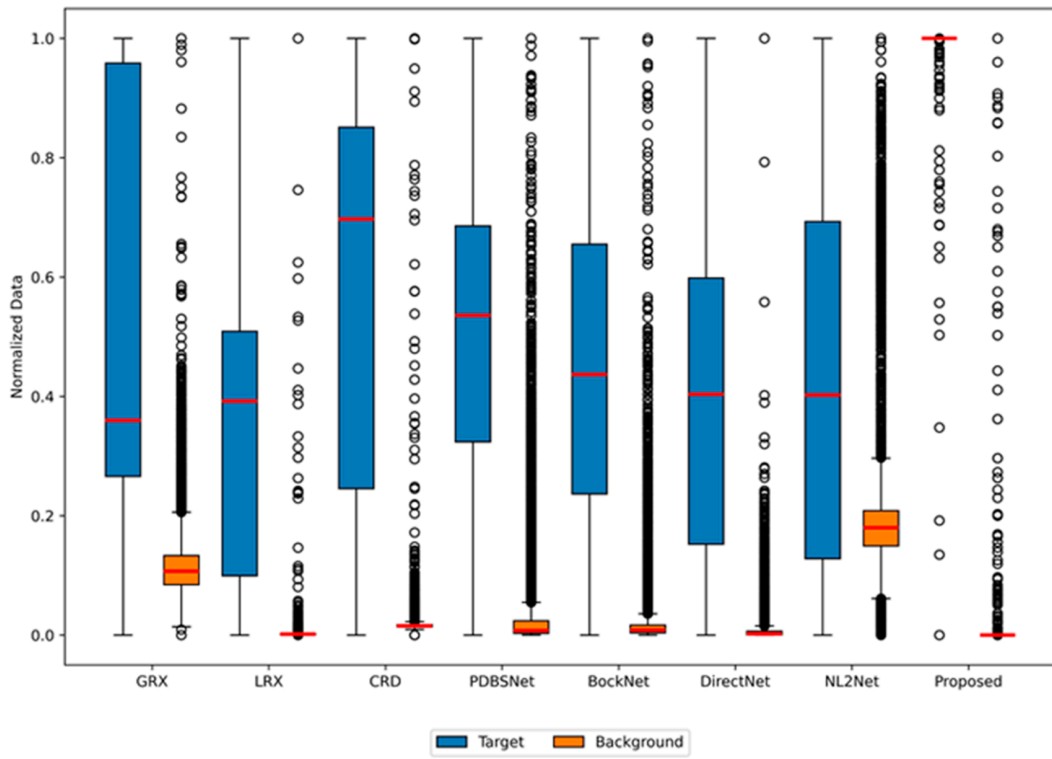

**Fig 16. Background-anomaly separation in Salinas-simulate dataset.**

a marginal gap of merely 0.0023, demonstrating near-state-of-the-art detection capability. Fig 17 demonstrated the performance comparison of different methods for hyperspectral anomaly detection using 2D and 3D ROC curves. In the 2D ROC curve $(P_D, P_F)$, the Bock-Net algorithm achieves the highest detection rate at equivalent false alarm rates, with its curve closest to the top-left corner. The proposed method follows closely, indicating strong performance. In the 2D ROC curve $(P_D, \tau)$, BockNet shows a significant increase in detection rate at high thresholds ($\tau \geq 10^{-2}$), while the proposed method also demonstrates a considerable rise, highlighting its effectiveness under stringent conditions. In the 2D ROC curve $(P_F, \tau)$, the proposed method maintains a lower false alarm rate across varying thresholds. The 3D ROC curve provides a comprehensive view, further emphasizing the proposed method's balanced performance in detection rate, false alarm rate, and threshold adaptability, placing

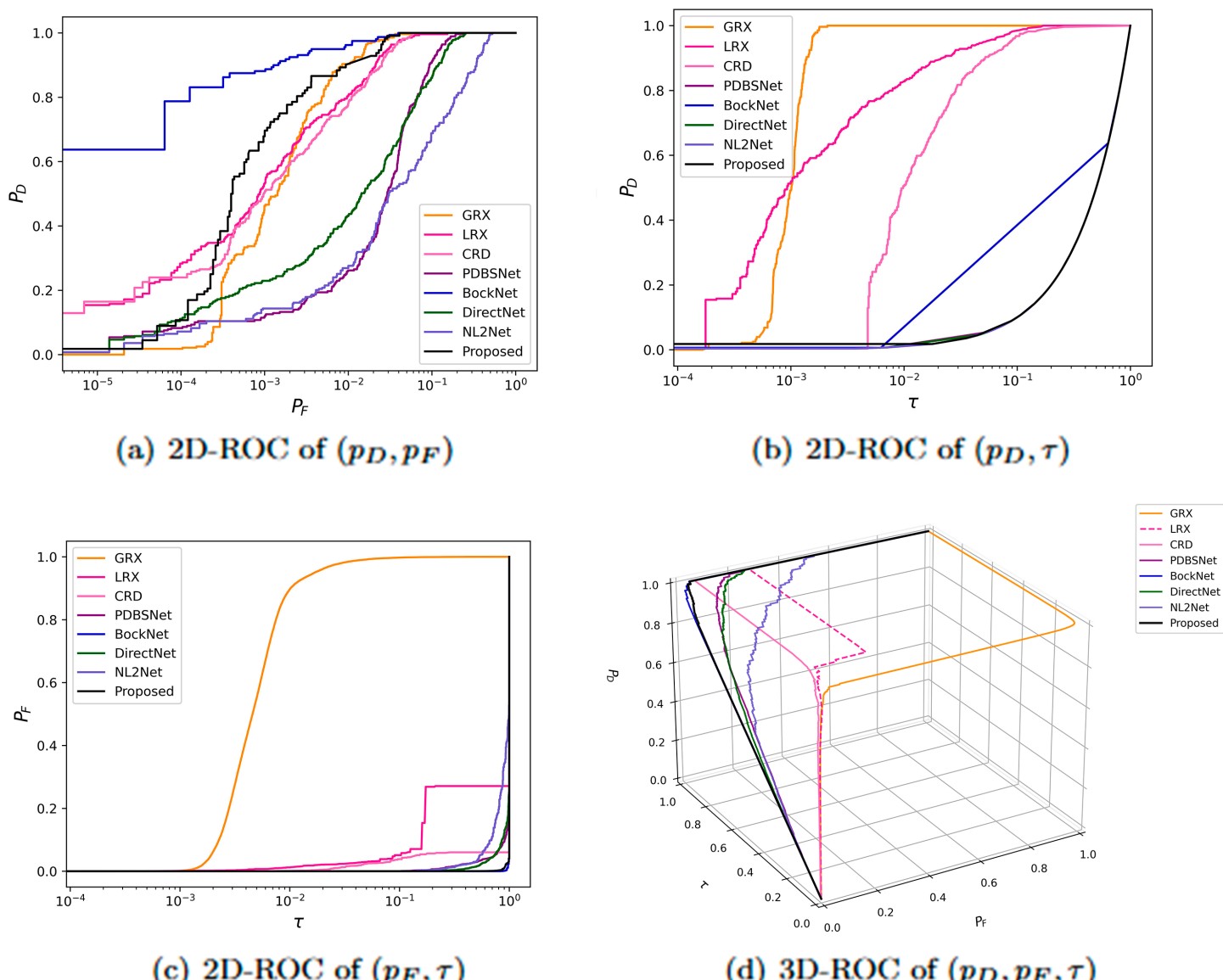

**Fig 17. Detection accuracy evaluation for the GrandIsle dataset.**

it as the second-best performer overall. Color maps for the anomaly detection displayed in Fig 18 revealed that the GRX, LRX, CRD,and DirectNet algorithms failed to detect anomalous targets. The PDBSNet, BockNet, and DirectNet algorithms mistakenly classified islands as anomalous targets. In contrast, the proposed algorithm precisely detected the positions and shapes of the anomalous targets, with the targets appearing bright and distinct, albeit with some minor noise. As shown in the boxplot (Fig 19), the proposed algorithm exhibits the largest gap between background and anomaly targets compared to other algorithms.

In experiments on the Cri dataset, the optimal dual-window configurations for LRX and CRD were determined as (25, 21) and (17, 13), respectively. As shown in Table 1, the proposed algorithm achieved the highest AUC value among all methods. Fig 20 demonstrated the performance comparison of different methods for hyperspectral anomaly detection using 2D and 3D ROC curves. In the 2D ROC curve $(P_D,P_F)$, the proposed method achieves the highest detection rate at equivalent false alarm rates, with its curve closest to the top-left corner. In the 2D ROC curve $(P_D,\tau)$, the proposed method shows a significant increase in detection rate at high thresholds $(\tau \geq 10^{-1})$, highlighting its effectiveness under stringent conditions. In the 2D ROC curve $(P_F,\tau)$, the proposed method maintains the lowest false alarm rate across varying thresholds. The 3D ROC curve further emphasizes the proposed method's balanced performance in detection rate, false alarm rate, and threshold adaptability. The anomaly detection results for the Cri dataset, visualized in Fig 21, reveal distinct performance variations among methods. While GRX, LRX, BockNet, and DirectNet partially localized anomalies, they failed to delineate target shapes clearly. CRD and NL2Net suffered from inadequate background suppression. In contrast, PDBSNet and the proposed algorithm precisely identified all anomalies in both location and morphology. Notably, the proposed method enhanced target-background contrast by sharply highlighting detected anomalies (rendered as bright

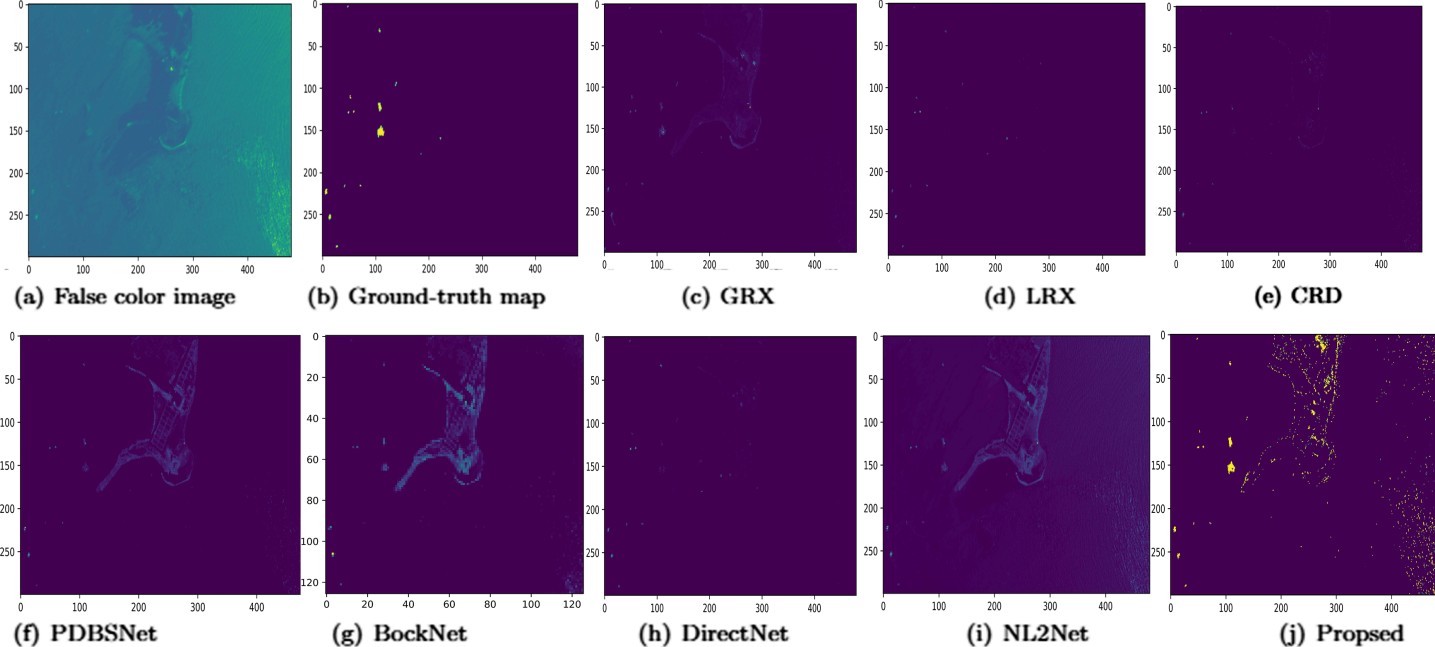

(a) False color image  (b) Ground-truth map  (c) GRX  (d) LRX  (e) CRD

(f) PDBSNet  (g) BockNet  (h) DirectNet  (i) NL2Net  (j) Propsed

**Fig 18. Detection maps obtained using different algorithms for the Grand Isle dataset.**

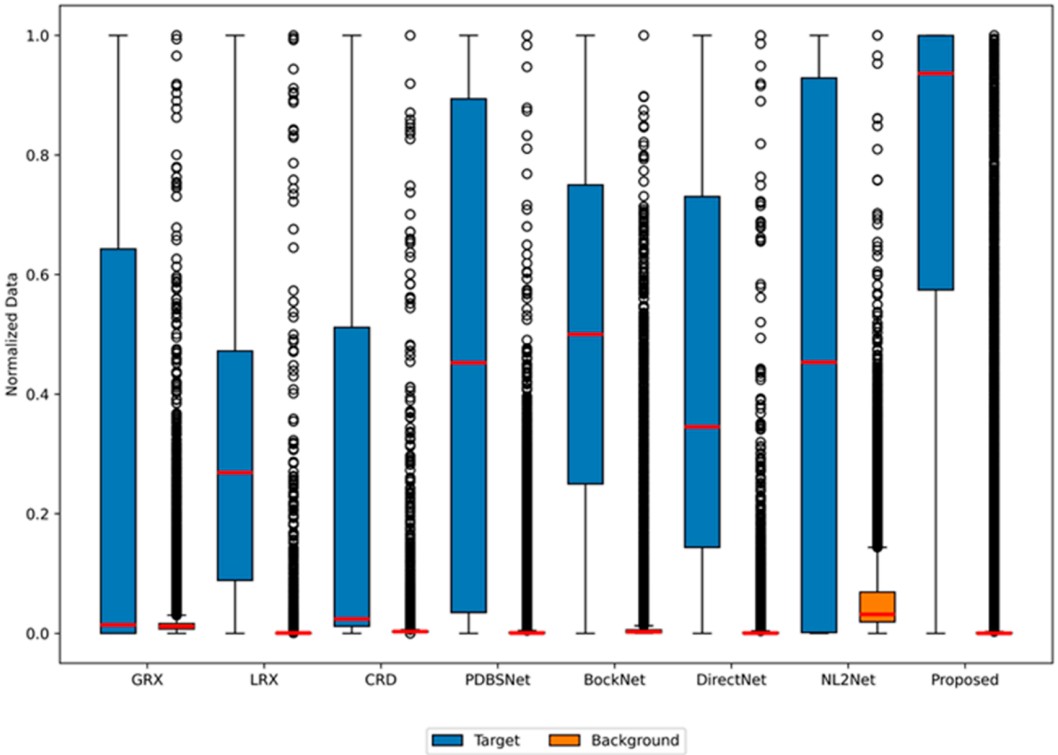

**Fig 19. Background-anomaly separation in GrandIsle dataset.**

regions), despite minor residual noise. As shown in the boxplot (Fig 22), the proposed algorithm exhibits the largest gap between background and anomaly targets compared to other algorithms.

## Computational complexity analysis

The computational complexities of the algorithms are as follows:

The GRX algorithm has a complexity of $O\left(n_1 n_2 d^2 + d^3\right)$.

The LRX algorithm has a complexity of $O(n_1 n_2 d^3 + [(\omega_{\text{out}}^2 - \omega_{\text{in}}^2) + 1]n_1 n_2 d^2 + n_1 n_2 d)$, where $\omega_{\text{out}}$ and $\omega_{in}$ are the sizes of the outer and inner windows, respectively.

The CRD algorithm has a complexity of $O(n_1 n_2(\omega_{\text{out}}^2 - \omega_{\text{in}}^2)d + (\omega_{\text{out}}^2 - \omega_{\text{in}}^2)^3)$.

The computational complexity of PDBSNet is determined by its core components. For a hyperspectral image of size $R \times C \times L$, the pixel-shuffle downsampling (PD) operation has time complexity $O(R \cdot C \cdot L \cdot s^2)$ with stride factor $s$. The dilated blind-spot network (DBSN) employs $N$ dilated convolution layers, each requiring $O(k^2 \cdot L^2 \cdot R \cdot C/s^2)$ computations for $k \times k$ kernels. The space complexity is dominated by intermediate feature maps at $O(L \cdot R \cdot C/s^2)$. Combined with the PD-inverse operation ($O(R \cdot C \cdot L \cdot s^2)$), the overall time complexity becomes $O\left(R \cdot C \cdot L \cdot s^2 + N \cdot k^2 \cdot L^2 \cdot R \cdot C/s^2\right)$, where the quadratic terms in $L$ and $N$ represent the primary computational bottlenecks.

The computational complexity of BockNet is primarily dominated by its simplified U-Net architecture, which involves multiple convolutional and pooling layers. For an input hyperspectral image of size $R \times R \times L$, the encoder and decoder components introduce $\mathcal{O}(R^2 LK)$

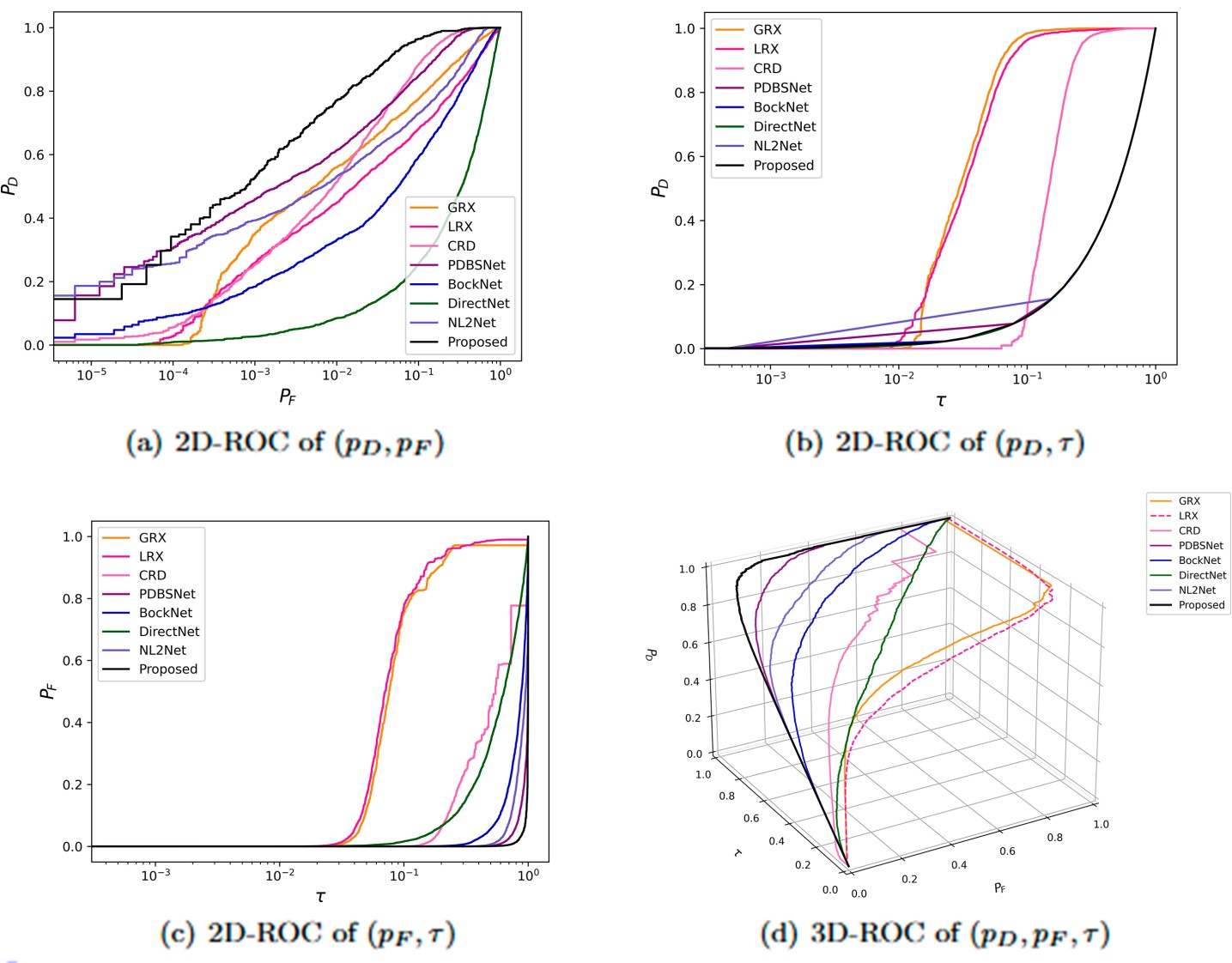

**Fig 20. Detection accuracy evaluation for the Cri dataset.**

operations, where $K$ represents the number of convolutional layers. The guard window mechanism requires rotating and fusing four directional feature maps, contributing an additional $\mathcal{O}(R^2L)$ term. Overall, the complexity scales as $\mathcal{O}(R^2LK)$, driven by the convolutional operations and linear in the number of spectral bands $L$ and quadratic in spatial dimensions $R$.

The computational complexity of DirectNet can be analyzed as follows. Given an input hyperspectral patch of size $W_{\text{out}} \times W_{\text{out}} \times L$ (where $W_{\text{out}}$ is the outer window size and $L$ is the spectral bands), the network depth $N_r = \frac{W_{\text{out}}-7}{4}$ grows linearly with window size. Each ResNet block contains two convolutional layers, leading to an overall complexity of $\mathcal{O}(N_r \cdot W_{\text{out}}^2 \cdot L \cdot C^2)$, where $C$ represents the number of feature channels. This simplifies to $\mathcal{O}(W_{\text{out}}^3 \cdot L)$ since $N_r \propto W_{\text{out}}$, making the complexity cubic in window size and linear in spectral dimensions.

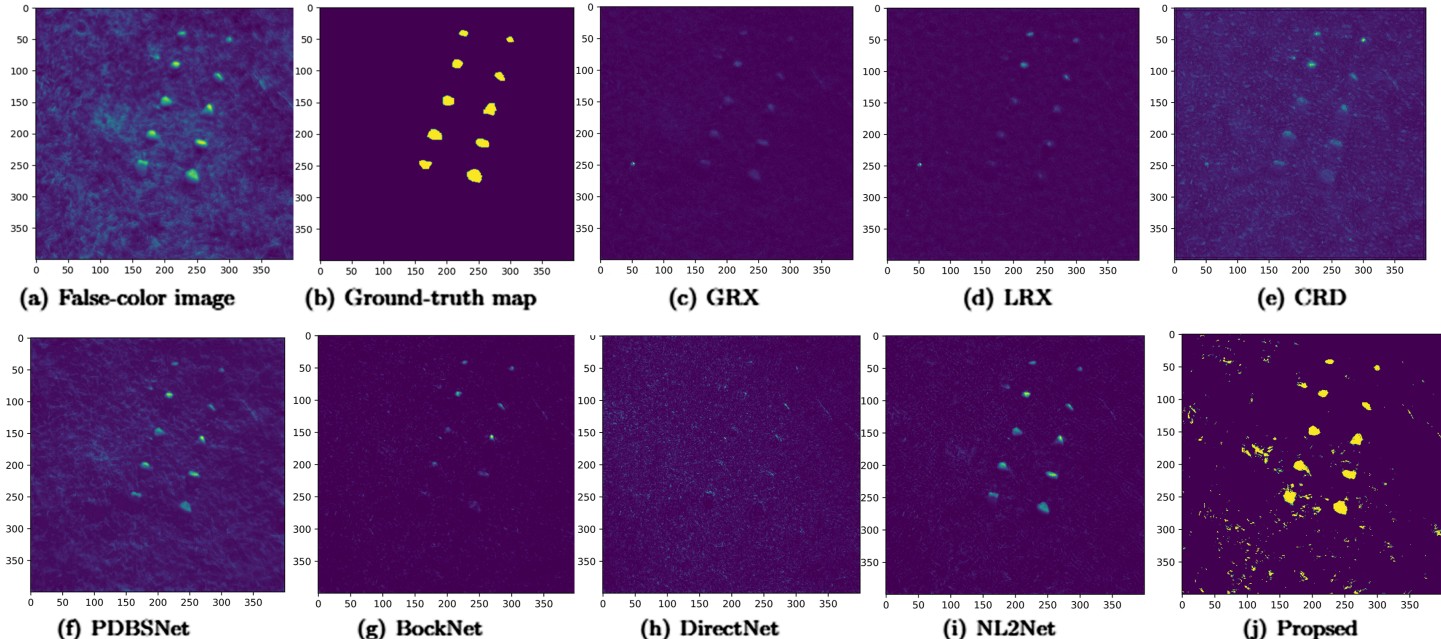

**Fig 21. Detection maps obtained using different algorithms for the Cri dataset.**

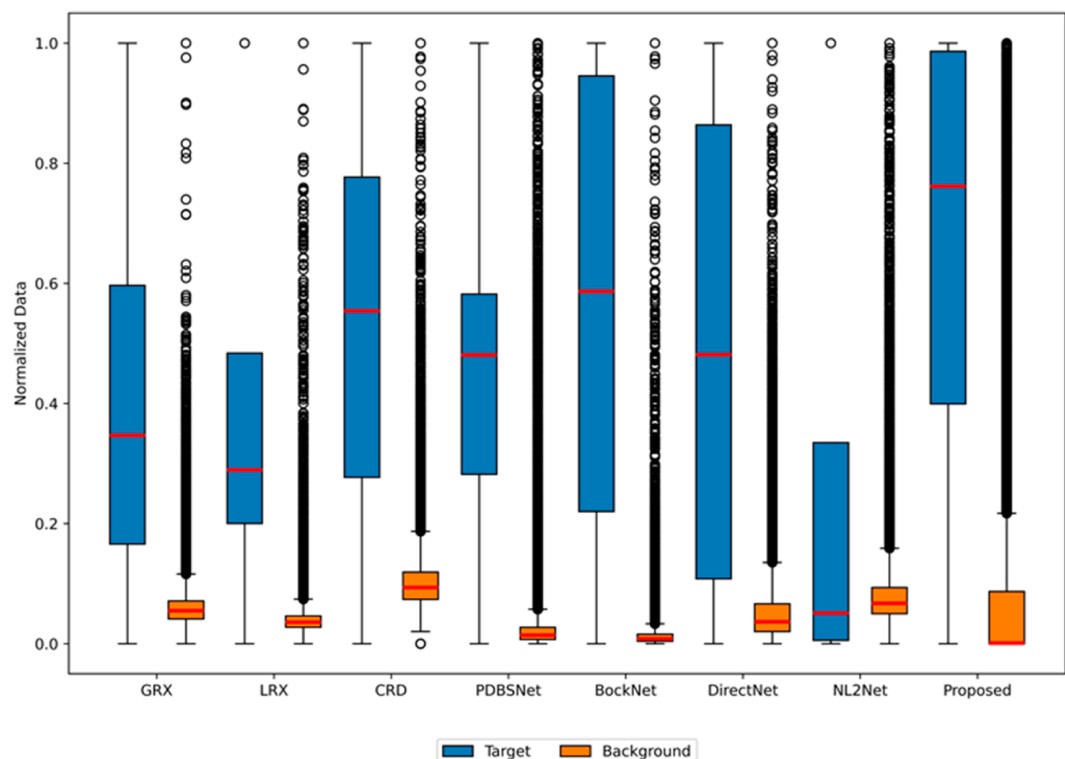

**Fig 22. Background-anomaly separation in Cri dataset.**

The computational complexity of NL2Net is dominated by its dual-branch architecture: for an input HSI cube with spatial size $H \times W$ and $C$ spectral bands, the local feature extraction branch (LFEB) incurs $O(HWk^2CD)$ complexity via $k \times k$ masked/dilated convolutions, while the nonlocal branch (NLFEB) reduces self-attention complexity from $O((HW)^2C)$ to $O(HWG^2C)$ through grid attention partitioning ($G \times G$ sub-regions). The pixel-shuffle downsampling with stride $F$ further scales spatial dimensions by $1/F^2$, yielding an overall complexity of $O\left(\frac{HW}{F^2}(k^2D + G^2)C\right)$.

Among them, the GAN–BWGNN HAD algorithm has the lowest computational complexity because the $\beta$ wavelet in GAN–BWGNN HAD is a polynomial, avoiding the Laplacian matrix decomposition and matrix multiplication, hence having a computational complexity of $O(kd^2 + kd + C|\mathcal{E}|)$. The algorithms runtimes are summarized in Table 2.

Based on the runtime comparisons in Table 2 , the proposed method achieves sub-second detection (0.20–0.28 s) on most datasets (AVIRIS-I/II, abu-beach-2/urban-3, Salinas-simulate), significantly outperforming traditional methods (LRX/CRD) by a speedup factor of 100–500 $\times$ (e.g., AVIRIS-I: 0.24 s vs. LRX 144.93 s) and deep learning models (NL2Net/BockNet) by 6–8 $\times$ (Salinas-simulate: 0.28 s vs. NL2Net 1.59 s). However, compared to PDBSNet, the proposed method exhibits marginally longer execution times on the AVIRIS-I (0.2408 s), AVIRIS-II (0.20479 s), abu-beach-2 (0.20179 s), and abu-urban-3 (0.20279 s) datasets, with an average increase of 0.04–0.05 s (corresponding to a 7.8%–10.5% difference) relative to PDBSNet's execution times (0.2178 s, 0.1628 s, 0.1581 s, and 0.1558 s, respectively). Notably, on the Cri dataset, the proposed method maintains real-time detection with a runtime of 1.606 s, demonstrating a three-order-of-magnitude speedup over CRD (1655 s). These results highlight the efficiency and scalability of the proposed approach across diverse hyperspectral imaging scenarios.

## Parameter analysis

1) Analysis of C:In the design framework of Beta Wavelets, the parameters $p$ and $q$ are constrained by $p+q = C$, where $C$ denotes the order of the wavelet kernel. This constraint ensures that all generated Beta wavelet kernels are $C$-th order polynomials, systematically covering the full spectral range from low to high frequencies. Specifically, the set of wavelet kernels $\{\mathcal{W}_{i,C-i}\}$, generated by iterating $i \in \{0, 1, ..., C\}$, comprises $C+1$ filters with complementary frequency characteristics. For instance, $\mathcal{W}_{0,C}$ functions as a low-pass filter primarily capturing low-frequency signals. As $i$ increases, the central frequency of the filter shifts toward higher frequencies, exhibiting band-pass properties. At $i = C$, $\mathcal{W}_{C,0}$ operates as a high-pass filter, focusing on high-frequency components. This multi-scale architecture enables flexible adaptation to anomaly features across diverse frequency bands.

**Table 2. Comparison of running time (seconds).**

| Dataset | GRX | LRX | CRD | PDBSNet | BockNet | DirectNet | NL2Net | Proposed |
|---|---|---|---|---|---|---|---|---|
| AVIRIS-I | 2.8122 | 144.9324 | 53.9466 | 0.2178 | 0.3658 | 0.3127 | 1.6423 | 0.2408 |
| AVIRIS-II | 1.9836 | 103.2134 | 49.2692 | 0.1628 | 0.2509 | 0.3057 | 1.6143 | 0.20479 |
| abu-beach-2 | 1.9086 | 73.836 | 14.1814 | 0.1581 | 0.2437 | 0.3035 | 1.5844 | 0.20179 |
| abu-urban-3 | 2.0099 | 114.5921 | 41.0893 | 0.1558 | 0.2537 | 0.3017 | 1.5904 | 0.20279 |
| Salinas-simulate | 3.6573 | 131.0882 | 128.0006 | 0.3037 | 0.8601 | 0.2917 | 1.5923 | 0.28171 |
| GrandIsle | 27.7151 | 924.3138 | 210.6745 | 2.1918 | 0.3206 | 0.6153 | 2.2187 | 2.59534 |
| Cri | 0.8182 | 319.2434 | 1655.3193 | 2.0413 | 3.1322 | 0.4185 | 2.9051 | 1.60635 |

**Proposition 1 (Spectral Locality).** Let $p > 0$, $q > 0$, and $X \sim \beta_{p,q}^*$, where $\beta_{p,q}^*$ is a band-pass distribution. The mean and variance of $X$ are:

$$\mu = \mathbb{E}(X) = \frac{2(p+1)}{p+q+2}$$

$$\sigma = \mathrm{Var}(X) = \frac{4(p+1)(q+1)}{(p+q+2)^2(p+q+3)}$$

When $p+q \to \infty$ under the condition $p = cq$, it is shown that $\sigma \to 0$ and $\mu = \frac{2c}{c+1}$, which can take any value in the interval $(0, 2)$.

To ensure $p$ and $q$ do not deviate significantly, $p = cq$ is employed. Proposition 4 is derived from the properties of the Beta distribution. It demonstrates that $\beta_{p,q}^*$ can concentrate on any $\mu \in (0, 2)$, indicating that Beta graph wavelets can be designed to target specific frequency bands for anomaly detection.

**Proposition 2 (Spatial Locality).** Let $v_i$ and $v_j$ be two nodes on a graph $\mathcal{G}$. The effect of a one-hot signal $\delta_i \in \mathbb{R}^N$ on node $v_j$ after the wavelet transform is denoted as $W_{p,q}\delta_i[j]$. It is observed that $W_{p,q}\delta_i$ is localized within $(p+q)$-hops of node $v_i$. Specifically, if the distance $d_G(v_i, v_j)$ exceeds $p + q$, then $W_{p,q}\delta_i[j]$ equals zero.

The results of Propositions 4 and 5 suggest that increasing the value of $C$ can enhance spectral locality, but this comes at the expense of reduced spatial locality. Conversely, decreasing $C$ improves spatial locality while potentially degrading spectral locality. This highlights a fundamental trade-off between the two domains.

To tackle the common "right-shift phenomenon" in spectral anomaly detection, where anomalies mainly appear in high-frequency bands, $C$ is usually set to values $\geq 2$ for adequate high-frequency coverage. The parameter $C$ is set by selecting the optimal value through experimental comparisons of AUC values across seven datasets, with $C \in \{2, 3, \ldots, 10\}$. The experimental results are shown in Fig 23 below.

The experimental results indicate that the optimal values of parameter $C$ for the datasets AVIRIS-I, AVIRIS-II, abu-beach-2, abu-urban-3, alinas-simulate, GrandIsle, and Cri are 2, 2, 2, 2, 3, 2, and 3, respectively.

2) Analysis of K: Since $K$ controls the number of node connected to each node, that is, it determines the edges of the graph, different values of $K$ will result in different graph structures. A too small value of K (such as $K < 10$) will make the model overly dependent on adjacent pixels, making it vulnerable to noise interference and difficult to capture long-distance correlations. A too large value of $K$ such as $K > 50$) will introduce redundant long-distance pixels, diluting the abnormal signals and increasing the computational complexity. In order to determine the optimal parameter, this study conducted a grid search for $K = 10, 20, 30, 40$, and $50$ on seven hyperspectral datasets and compared the AUC performance. The experiment is shown in Fig 24 below.

The experimental results indicate that the optimal values of parameter $C$ for the datasets AVIRIS-I, AVIRIS-II, abu-beach-2, abu-urban-3, alinas-simulate, GrandIsle, and Cri are 40, 20, 30, 10, 20, 50, and 50, respectively.

## Ablation study

Based on the ablation experimental results presented in Table 3, the proposed GAN-BWGNN algorithm demonstrates superior anomaly detection performance compared to the baseline BWGNN method (with the GAN module removed) across most hyperspectral datasets.

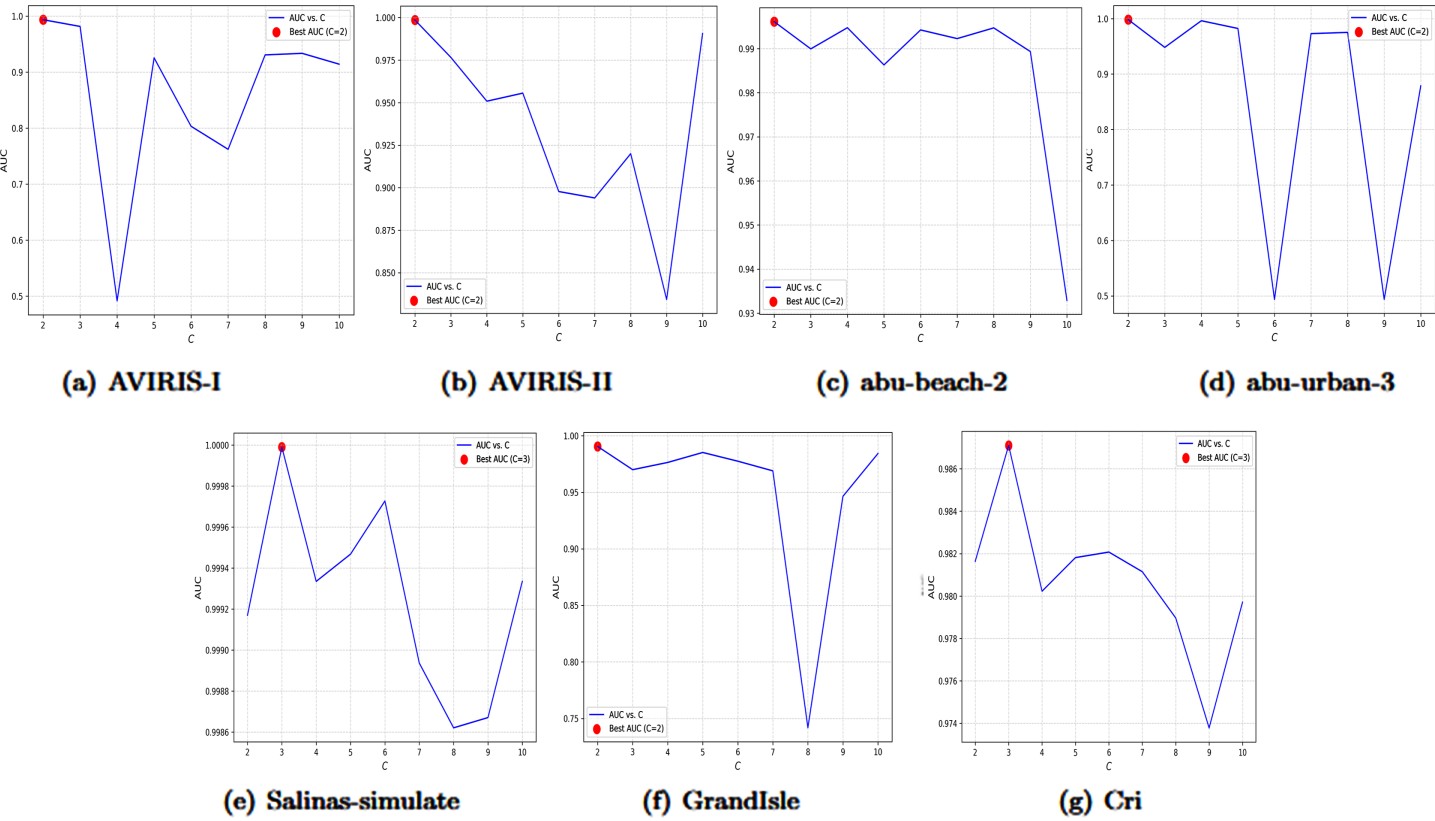

**Fig 23. Influence of C on the AUC on the seven datasets.**

Specifically, in typical scenarios such as abu-beach-2, abu-urban-3, and Grand Isle, the AUC values of GAN-BWGNN reach 0.9961, 0.9982, and 0.9966, respectively, representing relative improvements of 5.35%, 2.07%, and 2.22% over the baseline. This highlights the GAN module's effectiveness in enhancing the model's ability to discriminate spatial-spectral features between complex backgrounds and anomalous targets. Notably, both methods achieve saturated performance (0.9999) on the Salinas-simulate dataset, indicating lower detection difficulty in this simulated scenario, while only a 0.32% improvement is observed on the Cri dataset, potentially due to the strong spectral separability between anomalies and backgrounds in this scene. The experimental results conclusively validate that the Graph Attention Network significantly improves the robustness and generalization capability of hyperspectral anomaly detection through multi-scale neighborhood information fusion.

## Conclusions

We introduced the GAN–BWGNN HAD algorithm, which leverages the spatial context through GANs and addresses the spectral characteristics of anomalies using a BWGNN detector. Experimental results on five real-world datasets and one synthetic dataset demonstrated that GAN–BWGNN HAD had superior detection performance compared to the existing methods. The combination of the GAN and BWGNN allows for the utilization of both spatial and spectral information, providing a comprehensive analysis of hyperspectral data. The attention mechanism of the GAN enhances the adaptability of the proposed method to

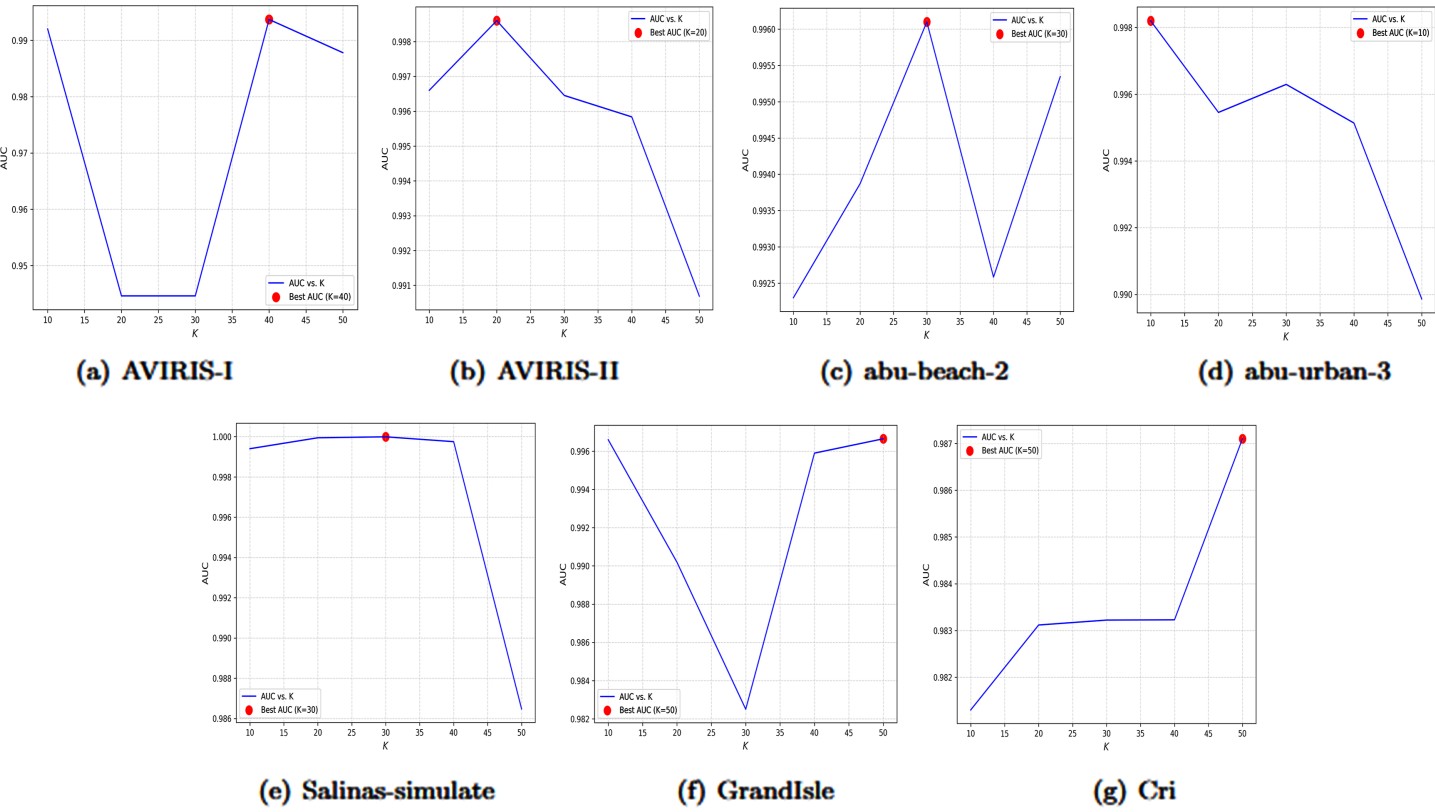

**Fig 24. Influence of K on the AUC on the seven datasets.**

**Table 3**. **AUC for ablation study.**

| Dataset | BWGNND | GAN–BWGNN |
|---|---|---|
| AVIRIS-I | 0.9916 | 0.9937 |
| AVIRIS-II | 0.9972 | 0.9986 |
| abu-beach-2 | 0.9426 | 0.9961 |
| abu-urban-3 | 0.9775 | 0.9982 |
| Salinas-simulate | 0.9999 | 0.9999 |
| GrandIsle | 0.9744 | 0.9966 |
| Cri | 0.9839 | 0.9871 |

various spatial contexts, whereas the beta wavelet-based spectral filtering via the BWGNN efficiently captures the right-shifted spectral energy associated with anomalies. Our findings show that the GAN–BWGNN HAD algorithm is a promising tool for HAD, offering improved accuracy and efficiency. Subsequent research will enhance this algorithm, apply it to additional remote sensing tasks, and investigate its potential integration with other sophisticated machine learning methodologies.

## Author contributions

**Conceptualization:** Ruhan A, Wenwen Feng.

**Data curation:** Ruhan A, Quanxue Gao, Siti Khadijah Ali.

**Formal analysis:** Ruhan A, Quanxue Gao, Xiaoni Zhang.

**Investigation:** Quanxue Gao, Xiaoni Zhang.

**Methodology:** Ruhan A.

**Project administration:** Xiaoni Zhang, Siti Khadijah Ali.

**Resources:** Xiaoni Zhang.

**Software:** Ruhan A.

**Supervision:** Xiaoni Zhang, Siti Khadijah Ali.

**Validation:** Ruhan A, Wenwen Feng.

**Visualization:** Ruhan A, Xiaoni Zhang, Wenwen Feng.

**Writing – original draft:** Ruhan A, Wenwen Feng.

**Writing – review & editing:** Xiaoni Zhang.

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
