## [Decision Letter · Decision Letter 0]

25 Mar 2025

PONE-D-25-02392Hyperspectral Anomaly Detection Leveraging Spatial Attention and Right-Shifted Spectral EnergyPLOS ONE

Dear Dr. A,

Thank you for submitting your manuscript to PLOS ONE. After careful consideration, we feel that it has merit but does not fully meet PLOS ONE’s publication criteria as it currently stands. Therefore, we invite you to submit a revised version of the manuscript that addresses the points raised during the review process.

Refine and condense the contributions section for clarity.Expand the literature review to include recent hyperspectral anomaly detection studies from 2024. Cite more recent contrastive learning methods to better highlight the novelty of the approach.Clarify how labeled data is obtained for training, particularly in Equation (17).Provide details on the anomaly detection process within the methodology section.Specify the structure of MLP and AGG functions (e.g., number of layers, activation functions, and whether AGG performs concatenation or summation).Justify parameter choices, including the k value in KNN and Beta wavelet parameters (a and b).Conduct ablation studies on the KNN parameter (k) and the impact of removing the graph attention network.Compare computation times of different methods, including inference times, for a comprehensive efficiency analysis.Expand evaluation metrics beyond AUC and ROC, incorporating 2D (PD,τ), 2D (PF, τ), and 3D-ROC curve comparisons.Discuss the limitations of the proposed algorithm, especially its lower performance on the AVIRIS-I dataset compared to CRD.

We look forward to receiving your revised manuscript.

Kind regards,

Panos Liatsis, PhD

Academic Editor

PLOS ONE

Journal Requirements:

3. Thank you for stating the following financial disclosure: [This research was funded by the Xi’an Peihua University Research Institutions and Innovation Team Special Project under Grant PHJT2406.]. 

4. In the online submission form, you indicated that [Data will be made available on request].

Reviewers' comments:

Reviewer's Responses to Questions

**Comments to the Author**

1. Is the manuscript technically sound, and do the data support the conclusions?

Reviewer #1: Yes

Reviewer #2: Yes

Reviewer #3: Partly

2. Has the statistical analysis been performed appropriately and rigorously? 

Reviewer #1: Yes

Reviewer #2: Yes

Reviewer #3: No

3. Have the authors made all data underlying the findings in their manuscript fully available?

Reviewer #1: Yes

Reviewer #2: Yes

Reviewer #3: No

4. Is the manuscript presented in an intelligible fashion and written in standard English?

Reviewer #1: Yes

Reviewer #2: Yes

Reviewer #3: No

5. Review Comments to the Author

Reviewer #1: This paper proposes a novel hyperspectral anomaly detection (HAD) algorithm, GAN–BWGNN HAD, which integrates a Graph Attention Network (GAN) and Beta Wavelet Graph Neural Network (BWGNN) to model spatial-spectral relationships. By leveraging graph structures and beta wavelets, the method enhances anomaly detection precision while improving computational efficiency. The suggestions can be considered:

1. The manuscript should ensure consistency in using full terms and abbreviations. Once an abbreviation (e.g., HAD) is introduced, it should be used consistently throughout the paper. A thorough revision is recommended.

2. The abstract should conclude with key experimental results to highlight the performance of the proposed method.

3. The introduction should begin with a brief overview of hyperspectral remote sensing and hyperspectral imaging to provide context for readers unfamiliar with the domain.

4. The contributions section lists multiple points; consider refining and consolidating them for clarity and conciseness.

5. The introduction should conclude with a brief summary of the paper’s structure to guide readers.

6. The proposed method appears to rely on labeled training data in Equation (17). The manuscript should clarify how labeled data is obtained for training.

7. The methodology section lacks a clear description of how anomalies are detected. The anomaly detection process should be explicitly detailed.

8. The experiment section lacks details about the Grand Isle dataset, including sensor specifications and data source information.

9. The manuscript should include a comparison of computation times for different methods to highlight efficiency improvements.

10. The introduction lacks discussion on recent hyperspectral anomaly detection approaches, particularly those based on convolutional neural network (Non-Local and Local Feature-Coupled Self-Supervised Network for Hyperspectral Anomaly Detection, Sliding Dual-Window-Inspired Reconstruction Network for Hyperspectral Anomaly Detection), low-rank and sparse representation (Hyperspectral Simultaneous Anomaly Detection and Denoising: Insights From Integrative Perspective), and generative adversarial networks (Frequency-to-spectrum mapping GAN for semisupervised hyperspectral anomaly detection).

Reviewer #2: In this manuscript, the authors propose a hyperspectral anomaly detection method that integrates spatial and spectral information by combining graph attention network with a Beta wavelet graph neural network. Please see my comments in the attached file.

Reviewer #3: Comments to PONE-D-25-02392

General Comment:

This paper proposes a novel method that integrates a Graph Attention Network with a Beta wavelet neural network, offering a new perspective on hyperspectral anomaly detection. The proposed method demonstrates promising results across multiple datasets and in comparison with several benchmark methods. However, there are certain shortcomings that need to be addressed. Therefore, major revisions are recommended. The detailed comments below outline specific areas for improvement.

Detail comments:

1. The literature review lacks recent studies on hyperspectral anomaly detection. The latest references only extend to 2023, with no citations from 2024. It is recommended to include more up-to-date studies.

2. The main contributions should be more concise. It is suggested to condense them into three key points.

3. The structure of the MLP and AGG functions in the network propagation process (Equations 12–16) is not clearly specified. Details such as the number of layers and activation functions in the MLP, as well as whether AGG performs concatenation or summation, should be provided.

4. There is a lack of discussion on the rationale behind parameter selection. The algorithm section should include a justification for the choice of the k value in KNN. Additionally, the specific parameter settings for the Beta wavelet (e.g., values of a and b) and their impact on the results should be elaborated.

5. Many of the contrastive learning methods cited in the paper are outdated. It is recommended to incorporate more recent methods from the past three years to better highlight the novelty and advantages of the proposed approach.

6. The evaluation metrics are too limited, as they only include AUC and ROC comparisons. Table 1, as well as Figures 3, 5, 7, 9, 11, 13, 15, and 17(a), all present the same AUC comparison, which increases the length of the paper without adding significant information. It is suggested to include 2D (PD,τ), 2D (PF, τ), and 3D-ROC curve comparisons to better illustrate the relationship between the detection rate and false alarm rate.

7. It is recommended to add boxplots to better visualize the separation between background and anomalies.

8. The paper claims that the Beta wavelet avoids Laplacian matrix decomposition (e.g., in Equation 9), yet the equations still involve U (the eigenvectors of the Laplacian matrix). The authors should clarify how the decomposition step is bypassed.

9. The paper lacks a comparison of inference times. It is recommended to include this aspect for a more comprehensive evaluation.

10. The limitations of the proposed algorithm are not sufficiently discussed. For instance, on the AVIRIS-I dataset, the algorithm performs worse than CRD. The authors should explain the possible reasons for this and suggest potential improvements.

6. PLOS authors have the option to publish the peer review history of their article (what does this mean?). If published, this will include your full peer review and any attached files.

Reviewer #1: No

Reviewer #2: No

Reviewer #3: No

---

## [Author Response · Author response to Decision Letter 1]

9 May 2025

Dear Reviewers,

Thank you for your thorough review and insightful comments on our manuscript. We have carefully considered each point and made the necessary revisions to improve the quality of our work.To facilitate your review, we have highlighted all the modified sections in yellow within the manuscript. We appreciate your time and effort in evaluating our paper and hope that our revisions meet your expectations.

Sincerely, Ruhan A

reviewer 1

This paper proposes a novel method that integrates a Graph Attention

Network with a Beta wavelet neural network, offering a new perspective

on hyperspectral anomaly detection. The proposed method demonstrates

promising results across multiple datasets and in comparison with several

benchmark methods. However, there are certain shortcomings that need to be addressed. Therefore, major revisions are recommended. The detailed comments below outline specific areas for improvement.

Detail comments:

1. The literature review lacks recent studies on hyperspectral anomaly

detection. The latest references only extend to 2023, with no citations

from 2024. It is recommended to include more up-to-date studies.

Respond Thank you for your suggestion. We have added several recent studies from 2024 and 2025 to ensure the literature review is comprehensive and up-to-date.

2. The main contributions should be more concise. It is suggested to

condense them into three key points.

Respond We sincerely appreciate your suggestion to streamline the main contributions of our work. In response, we have condensed them into three key points to enhance clarity and focus. Thank you for your valuable feedback.

3. The structure of the MLP and AGG functions in the network

propagation process (Equations 12–16) is not clearly specified. Details

such as the number of layers and activation functions in the MLP, as

well as whether AGG performs concatenation or summation, should be

provided.

Respond Thank you for your valuable feedback regarding the structure of the MLP and AGG functions in our network propagation process. We sincerely appreciate your suggestions and have addressed them by providing detailed specifications in the "Experiments" section.

4.There is a lack of discussion on the rationale behind parameter selection. The algorithm section should include a justification for the choice of the k value in KNN. Additionally, the specific parameter settings for the Beta wavelet (e.g., values of a and b) and their impact on the results should be elaborated.

Respond Thank you for your valuable feedback. Following your suggestions, we have added a "Parameter Analysis" section to the paper, discussing the selection of k values in KNN and the settings of Beta wavelet parameters (such as a and b) and their impact on the results. We hope these additions meet your expectations.

5. Many of the contrastive learning methods cited in the paper are

outdated. It is recommended to incorporate more recent methods from

the past three years to better highlight the novelty and advantages of

the proposed approach.

Respond We sincerely appreciate your insightful comments regarding the cited contrastive methods. To address your suggestion, we have incorporated more recent methods into our comparisons, including PDBSNet (2023), BockNet (2023), DirectNet (2024), and NL2Net (2025), into our comparative analysis. Thank you again for your valuable feedback.

6. The evaluation metrics are too limited, as they only include AUC and

ROC comparisons. Table 1, as well as Figures 3, 5, 7, 9, 11, 13, 15, and

17(a), all present the same AUC comparison, which increases the

length of the paper without adding significant information. It is

suggested to include 2D (PD,τ), 2D (PF, τ), and 3D-ROC curve

comparisons to better illustrate the relationship between the detection

rate and false alarm rate.

Respond We sincerely appreciate your feedback regarding the evaluation metrics. To address your suggestion, we have removed the redundant figures (3, 5, 7, 9, 11, 13, 15, and 17(a)) and added comparisons using 2D (PD,τ), 2D (PF, τ), and 3D-ROC curves. These new metrics are now presented in Figures 2, 5, 8, 11, 14, 17, and 20 to better illustrate the relationship between detection rate and false alarm rate. Thank you again for your valuable comments.

7. It is recommended to add boxplots to better visualize the separation

between background and anomalies.

Respond We sincerely appreciate your suggestion to enhance the visualization of the separation between background and anomalies. In response, we have added boxplots to Figures 4, 7, 10, 13, 16, 19, and 22. Thank you for your valuable feedback.

8. The paper claims that the Beta wavelet avoids Laplacian matrix

decomposition (e.g., in Equation 9), yet the equations still involve U

(the eigenvectors of the Laplacian matrix). The authors should clarify

how the decomposition step is bypassed.

Respond Thank you very much for your valuable comments and suggestions regarding our paper. You raised an important point about the Beta Wavelet avoiding the decomposition of the Laplacian matrix while the equations still involve (the eigenvectors of the Laplacian matrix). We are more than happy to provide further clarification.

In theory, the Beta Wavelet transform is indeed expressed as, whererepresents the eigenvectors of the Laplacian matrix L. However, in practice, explicit computation of U is not required. The key insight is that the Beta Wavelet kernel is designed as a polynomial function of the Laplacian matrix L, which allows us to perform computations directly in the spatial domain without the need for explicit eigendecomposition. More specifically, the Beta Wavelet kernel is formulated as:

By recursively computing the powers of L (e.g., and ), we can efficiently implement the polynomial kernel, thus avoiding the high computational cost associated with explicit eigendecomposition.

We understand that the presence of U in the equations might cause confusion, as it is included for theoretical completeness. To address this, we have revised the methodology section of the paper to provide a more detailed explanation of how the Beta Wavelet transform is achieved through polynomial approximations without explicitly computing the eigenvectors matrix U.

Once again, we sincerely appreciate your insightful feedback and are committed to improving our paper based on your suggestions.

9. The paper lacks a comparison of inference times. It is recommended to

include this aspect for a more comprehensive evaluation.

Respond We sincerely appreciate your suggestion regarding the comparison of inference times. In response, we have included this aspect in the "Computational Complexity Analysis" section. Thank you for your valuable feedback.

10.The limitations of the proposed algorithm are not sufficiently discussed. For instance, on the AVIRIS-I dataset, the algorithm performs worse than CRD. The authors should explain the possible reasons for this and suggest potential improvements.

Respond We sincerely appreciate your feedback regarding the limitations of our proposed algorithm. In response, we have added a discussion on the possible reasons for the algorithm's suboptimal performance on the AVIRIS-I dataset. Thank you for your valuable comments.

reviewer 2

In this manuscript, the authors propose a hyperspectral anomaly detection method that integrates spatial and spectral information by combining graph attention network with a Beta wavelet graph neural network. Below are some suggestions aimed at further enhancing the quality and readability of the manuscript:

1.Originality of Figures: In the section titled "Right-Shift Phenomenon", Figure 1 appears highly similar to a figure in the published article "Hyperspectral Anomaly Detection Based on a Beta Wavelet Graph Neural Network" (IEEE Multimedia, vol. 31, no. 2, pp. 69–79, Apr.–Jun. 2024). To ensure the originality of the manuscript, it is recommended that the authors make necessary modifications to Figure 1 or provide a more detailed explanation to distinguish it from the existing work.

Respond Thank you for your careful review and valuable feedback on the manuscript's figure originality. The article “Hyperspectral Anomaly Detection Based on a Beta Wavelet Graph Neural Network” you mentioned is my previous work. Figure 1 was an example of the right - shift phenomenon. To ensure originality and avoid content repetition, I've removed Figure 1. Thanks again for your constructive criticism.

2.Algorithm Pseudocode: Although the authors have provided a thorough description of the algorithm, to facilitate readers' quick understanding of the algorithm's workflow, it is suggested that a complete and clear pseudocode flowchart with standardized notation be added. This would significantly enhance the readability and reproducibility of the manuscript.

Respond Thank you for your insightful suggestion. I fully recognize the importance of pseudocode in enhancing the readability and reproducibility of the manuscript. In response to your feedback, I have added a complete and clear pseudocode flowchart with standardized notation. Once again, I appreciate your constructive criticism, which has been invaluable in improving the quality of our work.

3.Label Acquisition: Based on the description of Equation (17), the proposed method appears to be a supervised method. However, obtaining labels for hyperspectral anomaly detection is typically challenging. The authors are encouraged to elaborate on how these labels were acquired, specifically whether the ground truth maps of the data were used for training. This would help readers better understand the feasibility and practicality of the method.

Respond Thank you for your valuable feedback. We acknowledge the challenges of label acquisition in hyperspectral anomaly detection. In this work, labels were derived from standard ground truth maps of publicly available hyperspectral datasets. For training, only 40% of the labeled data was utilized to avoid overfitting, with the remaining 60% acting as unlabeled samples for anomaly score refinement. We thank you for your suggestion and have added a dedicated section in the Experiments part to detail the data sources and partitioning.

4.Ablation Experiments: The manuscript lacks comparative experiments on the effects of the KNN parameter k and the performance without the graph attention network. It is recommended that the authors supplement relevant ablation experiments to more comprehensively evaluate the performance of the proposed method.

Respond Thank you for your feedback. We have added a "Parameter Analysis" section discussing the k value in KNN and an "Ablation Study" section comparing the performance with and without the graph attention network. We hope these additions address your concerns.

5. Computational Complexity Analysis: In the "Computational Complexity Analysis" section, the authors have provided a detailed analysis of the computational complexity of various algorithms. However, directly presenting the actual runtime of each algorithm on the dataset would offer a more intuitive understanding of the performance of the proposed method.

Respond We sincerely appreciate your suggestion regarding the comparison of inference times. In response, we have included this aspect in the "Computational Complexity Analysis" section. Thank you for your valuable feedback.

6.Spelling Error Correction: The authors are advised to carefully check the spelling throughout the manuscript. Specifically, on page 2, line 35 of the Introduction section, there is a noticeable spelling error ("Itt"), which should be corrected.

Respond� We sincerely apologize for the oversight and confirm that the spelling error on page 2, line 35 ("Itt") has been corrected to "It." Thank you for your vigilance.

7. Some new works highly related to this topic are suggested to be referred, such as:

HADGSM: A unified nonconvex framework for hyperspectral anomaly detection. IEEE Transactions on Geoscience and Remote Sensing, 2024, 62, 5503415.

BS3LNet: A new blind-spot self-supervised learning network for hyperspectral anomaly detection. IEEE Transactions on Geoscience and Remote Sensing, 2023, 61, 5504218.

Hyperspectral anomaly detection based on chessboard topology. IEEE Transactions on Geoscience and Remote Sensing, 2023, 61, 5505016.

Information entropy estimation based on point-set topology for hyperspectral anomaly detection. IEEE Transactions on Geoscience and Remote Sensing, 2024, 62, 5523415.

Respond Thank you for bringing these recent works to our attention. We have incorporated these references into the revised manuscript to ensure a comprehensive review of the latest advancements in hyperspectral anomaly detection.

---

## [Decision Letter · Decision Letter 1]

9 Jun 2025

PONE-D-25-02392R1Hyperspectral Anomaly Detection Leveraging Spatial Attention and Right-Shifted Spectral EnergyPLOS ONE

Dear Dr. A,

Thank you for submitting your manuscript to PLOS ONE. After careful consideration, we feel that it has merit but does not fully meet PLOS ONE’s publication criteria as it currently stands. Therefore, we invite you to submit a revised version of the manuscript that addresses the points raised during the review process.

Please improve manuscript organization and readability by streamlining the literature review, providing clear context for readers new to the field, and ensuring the structure of the introduction and abstract effectively highlight the motivation, contributions, and results of the work.Please be consistent in the use of abbreviations, terminology, and formatting throughout the manuscript. Provide an improved explanation of the experimental setup, including dataset details, data sources, and the process for obtaining labeled data. The presentation of results should be enhanced, especially in the abstract, to better showcase the method’s performance.The manuscript should be carefully reviewed for technical errors, such as figure reference issues, incomplete figures, and duplicate citations. All visual and reference elements should be complete, accurate, and properly formatted.The related work section should be updated to reflect recent advances in hyperspectral anomaly detection, particularly with respect to neural network-based and generative approaches, to better position the proposed method within the current research landscape.

We look forward to receiving your revised manuscript.

Kind regards,

Panos Liatsis, PhD

Academic Editor

PLOS ONE

Reviewers' comments:

Reviewer's Responses to Questions

**Comments to the Author**

1. If the authors have adequately addressed your comments raised in a previous round of review and you feel that this manuscript is now acceptable for publication, you may indicate that here to bypass the “Comments to the Author” section, enter your conflict of interest statement in the “Confidential to Editor” section, and submit your "Accept" recommendation.

Reviewer #1: All comments have been addressed

Reviewer #3: (No Response)

2. Is the manuscript technically sound, and do the data support the conclusions?

Reviewer #1: Yes

Reviewer #3: (No Response)

3. Has the statistical analysis been performed appropriately and rigorously? 

Reviewer #1: Yes

Reviewer #3: (No Response)

4. Have the authors made all data underlying the findings in their manuscript fully available?

Reviewer #1: Yes

Reviewer #3: (No Response)

5. Is the manuscript presented in an intelligible fashion and written in standard English?

Reviewer #1: Yes

Reviewer #3: (No Response)

6. Review Comments to the Author

Reviewer #1: This paper proposes a novel hyperspectral anomaly detection (HAD) algorithm, GAN–BWGNN HAD, which integrates a Graph Attention Network (GAN) and Beta Wavelet Graph Neural Network (BWGNN) to model spatial-spectral relationships. By leveraging graph structures and beta wavelets, the method enhances anomaly detection precision while improving computational efficiency. The suggestions can be considered:

1. The abstract should conclude with key experimental results to highlight the performance of the proposed method.

2. The introduction should begin with a brief overview of hyperspectral remote sensing and hyperspectral imaging to provide context for readers unfamiliar with the domain.

3. The introduction should conclude with a brief summary of the paper’s structure to guide readers.

4. The proposed method appears to rely on labeled training data in Equation (17). The manuscript should clarify how labeled data is obtained for training.

5. The experiment section lacks details about the Grand Isle dataset, including sensor specifications and data source information.

6. References 1 and 27 are duplicated.

7. The introduction lacks discussion on recent hyperspectral anomaly detection approaches, particularly those based on convolutional neural network (Global Feature-Injected Blind-Spot Network for Hyperspectral Anomaly Detection), low-rank and sparse representation (Hyperspectral Simultaneous Anomaly Detection and Denoising: Insights From Integrative Perspective), and generative adversarial networks (Frequency-to-spectrum mapping GAN for semisupervised hyperspectral anomaly detection).

Reviewer #3: (No Response)

7. PLOS authors have the option to publish the peer review history of their article (what does this mean?). If published, this will include your full peer review and any attached files.

Reviewer #1: No

Reviewer #3: No

---

## [Author Response · Author response to Decision Letter 2]

11 Jul 2025

Dear Editors and Reviewers,

We sincerely appreciate your valuable time and constructive comments on our manuscript. All suggestions have been systematically addressed in the revised manuscript. Responses to your specific comments are provided in blue text below, while modifications in the manuscript are highlighted in yellow for easy tracking. These revisions significantly enhance the clarity and rigor of our work.

Sincerely, Ruhan A

Reviewer 1: This paper proposes a novel hyperspectral anomaly detection (HAD) algorithm, GAN–BWGNN HAD, which integrates a Graph Attention Network (GAN) and Beta Wavelet Graph Neural Network (BWGNN) to model spatial-spectral relationships. By leveraging graph structures and beta wavelets, the method enhances anomaly detection precision while improving computational efficiency. The suggestions can be considered:

1. The abstract should conclude with key experimental results to highlight the performance of the proposed method.

Response: Thank you for your valuable suggestion. We have revised the abstract to include the key experimental results, as recommended, to better highlight the performance of our method. We appreciate your insightful feedback.

2. The introduction should begin with a brief overview of hyperspectral remote sensing and hyperspectral imaging to provide context for readers unfamiliar with the domain.

Response:Thank you for your thoughtful suggestion. We have revised the introduction to include a brief overview of hyperspectral remote sensing and imaging, providing essential context for readers unfamiliar with the domain. Your feedback greatly improved the accessibility of our work.

3. The introduction should conclude with a brief summary of the paper’s structure to guide readers.

Response:Thank you for your valuable suggestion. We have revised the introduction to conclude with a brief summary of the paper's structure, as recommended, to guide readers through the subsequent sections. Your feedback greatly enhanced the clarity of our manuscript.

4. The proposed method appears to rely on labeled training data in Equation (17).The manuscript should clarify how labeled data is obtained for training.

Response: Thank you for your valuable suggestion. We have explicitly clarified how labeled training data is obtained in the "Datasets" section as recommended. Your feedback significantly strengthened our methodology description.

5. The experiment section lacks details about the Grand Isle dataset, including sensor specifications and data source information.

Response:Thank you for your insightful suggestion. We have supplemented the experiment section with Grand Isle dataset details (including sensor specifications and data sourcing).

6. References 1 and 27 are duplicated.

Response:Thank you for identifying the duplication in References 1 and 27. We sincerely apologize for this oversight and have fully revised the reference list to correct the error. Your meticulous review is greatly appreciated.

7. The introduction lacks discussion on recent hyperspectral anomaly detection approaches, particularly those based on convolutional neural network (Global Feature-Injected Blind-Spot Network for Hyperspectral Anomaly Detection), low-rank and sparse representation (Hyperspectral Simultaneous Anomaly Detection and Denoising: Insights From Integrative Perspective), and generative adversarial networks (Frequency-to-spectrum mapping GAN for semisupervised hyperspectral anomaly detection).

Response:Thank you for highlighting this critical gap. We have supplemented the introduction with discussions of recent hyperspectral anomaly detection approaches, including convolutional neural network (Global Feature-Injected Blind-Spot Network for Hyperspectral Anomaly Detection), low-rank and sparse representation (Hyperspectral Simultaneous Anomaly Detection and Denoising: Insights From Integrative Perspective), and generative adversarial networks (Frequency-to-spectrum mapping GAN for semisupervised hyperspectral anomaly detection). Your highly pertinent suggestions significantly strengthened the survey's comprehensiveness.

Reviewer 2:

1.The manuscript exhibits inconsistent and non-standard use of abbreviations. Terms such as hyperspectral anomaly detection (HAD) and graph attention network(GAN) are repeatedly redefined throughout the paper, which is not acceptable in academic writing. Abbreviations should be introduced only once—upon first use—and used consistently thereafter. The authors are encouraged to revise the manuscript thoroughly to ensure proper and standardized use of abbreviations.

Response:We sincerely apologize for the oversight in abbreviation usage. A systematic review of all abbreviations has been conducted throughout the manuscript to ensure standardized definition upon first occurrence (e.g., HAD, GAN) and consistent application thereafter, aligning fully with academic writing standards.

2. The literature review section in the Introduction is overly detailed and lacks conciseness. It is recommended that the authors streamline this section by emphasizing the key techniques, their limitations, and how the proposed method addresses existing gaps, rather than providing lengthy explanations of each related work.

Response:Thank you for this valuable suggestion to enhance conciseness. We have thoroughly streamlined the literature review in the Introduction by focusing on core techniques, their limitations, and our method's contributions to address existing gaps, significantly improving the section's clarity.

3.On page 15, line 416, the manuscript refers to ". Figure ??", which suggests a citation or rendering error. All figure references should be correctly formatted and verified.

Response: We sincerely apologize for the oversight in line 416 and have corrected the figure reference. All figure citations and references have been validated for correct formatting.

4.Figure 8 appears to be incomplete or missing content in the current version of the manuscript.

Response: We sincerely apologize for the oversight in Figure 8. This figure has been fully revised to complete its missing content. Thank you for your meticulous review – we have systematically verified all figures to enhance manuscript clarity.

5. The phrase “the Proposed method” appears multiple times with incorrect

capitalization. Please revise all instances to “the proposed method”.

Response:Thank you for catching the capitalization inconsistency. We have systematically revised all instances of "the Proposed method" to lowercase ("the proposed method") throughout the manuscript to ensure grammatical precision.

6.In the first paragraph of the Computational Complexity Analysis section, there are several occurrences of inconsistent or excessive spacing between words.

Response:Thank you for highlighting the spacing inconsistencies in the Computational Complexity Analysis section. We have standardized all word spacing throughout this paragraph to align with journal formatting guidelines, ensuring uniform typography.

---

## [Decision Letter · Decision Letter 2]

5 Aug 2025

Hyperspectral Anomaly Detection Leveraging Spatial Attention and Right-Shifted Spectral Energy

PONE-D-25-02392R2

Dear Dr. A,

We’re pleased to inform you that your manuscript has been judged scientifically suitable for publication and will be formally accepted for publication once it meets all outstanding technical requirements.

Kind regards,

Panos Liatsis, PhD

Academic Editor

PLOS ONE

Additional Editor Comments (optional):

Reviewers' comments:

Reviewer's Responses to Questions

**Comments to the Author**

1. If the authors have adequately addressed your comments raised in a previous round of review and you feel that this manuscript is now acceptable for publication, you may indicate that here to bypass the “Comments to the Author” section, enter your conflict of interest statement in the “Confidential to Editor” section, and submit your "Accept" recommendation.

Reviewer #1: All comments have been addressed

Reviewer #3: (No Response)

2. Is the manuscript technically sound, and do the data support the conclusions?

Reviewer #1: Yes

Reviewer #3: (No Response)

3. Has the statistical analysis been performed appropriately and rigorously? 

Reviewer #1: Yes

Reviewer #3: (No Response)

4. Have the authors made all data underlying the findings in their manuscript fully available?

Reviewer #1: Yes

Reviewer #3: (No Response)

5. Is the manuscript presented in an intelligible fashion and written in standard English?

Reviewer #1: Yes

Reviewer #3: (No Response)

6. Review Comments to the Author

Reviewer #1: The authors have revised the manuscript about hyperspectral anomaly detection, and I have no more comments. The paper is acceptable.

Reviewer #3: (No Response)

7. PLOS authors have the option to publish the peer review history of their article (what does this mean?). If published, this will include your full peer review and any attached files.

Reviewer #1: No

Reviewer #3: No

---

## [Editor Report · Acceptance letter]

PONE-D-25-02392R2

PLOS ONE

Dear Dr. A,

I'm pleased to inform you that your manuscript has been deemed suitable for publication in PLOS ONE. Congratulations! Your manuscript is now being handed over to our production team.

Kind regards,

on behalf of

Professor Panos Liatsis

Academic Editor

PLOS ONE